# Rethinking Reward Model Evaluation: Are We Barking up the Wrong Tree?

**Xueru Wen**[1,2*]   **Jie Lou**[3†]   **Yaojie Lu**[1†]   **Hongyu Lin**[1]   **Xing Yu**[3]   **Xinyu Lu**[1,2]
**Ben He**[2]   **Xianpei Han**[1]   **Debing Zhang**[3]   **Le Sun**[1]

[1] Chinese Information Processing Laboratory, Institute of Software, Chinese Academy of Sciences
[2] University of Chinese Academy of Sciences
[3] Xiaohongshu Inc

{wenxueru2022,luxinyu2021}@iscas.ac.cn
{luyaojie,hongyu,xianpei,sunle}@iscas.ac.cn
{benhe}@ucas.ac.cn
{loujie0822}@gmail.com
{dengyang}@xiaohongshu.com

## Abstract

Reward Models (RMs) are crucial for aligning language models with human preferences. Currently, the evaluation of RMs depends on measuring accuracy against a validation set of manually annotated preference data. Although this method is straightforward and widely adopted, the relationship between RM accuracy and downstream policy performance remains under-explored. In this work, we conduct experiments in a synthetic setting to investigate how differences in RM measured by accuracy translate into gaps in optimized policy performance. Our findings reveal that while there is a weak positive correlation between accuracy and downstream performance, policies optimized towards RMs with similar accuracy can exhibit quite different performance. Moreover, we discover that the way of measuring accuracy significantly impacts its ability to predict the final policy performance. Through the lens of the Regressional Goodhart effect, we recognize that accuracy, when used for measuring RM quality, can fail to fully capture the potential RM overoptimization. This underscores the inadequacy of relying solely on accuracy to reflect their impact on policy optimization.

## 1 Introduction

Reinforcement Learning from Human Feedback (RLHF) (Ibarz et al., 2018; Ouyang et al., 2022) has emerged as a prominent paradigm for aligning Large Language Models (Yang et al., 2024; Dubey et al., 2024). The reward model (Ng & Russell, 2000; Brown & Niekum, 2019; Palan et al., 2019) plays a crucial role in this process by substituting human preferences for model optimization. However, building an RM that fully captures human preferences is highly challenging (Armstrong & Mindermann, 2019; Skalse & Abate, 2023; Lambert et al., 2023). Therefore, the RM can be an imperfect proxy for ideal preferences and cause downstream performance deterioration when optimized against it, known as reward model overoptimization (Gao et al., 2022). This phenomenon, as a result of Goodhart's law (Karwowski et al., 2023), presents a critical challenge to the RLHF.

The difficulty of constructing an ideal RM require a reliable evaluation to capture potential negative impacts in policy optimization. To date, common practice for evaluating the RM includes directly assessing the optimized policy (Hendrycks et al., 2021; Li et al., 2023) and computing accuracy on a fixed dataset (Lambert et al., 2024). While the former approach serves as a final metric, it is constrained by the cost of optimization and cannot distinguish whether undesirable behaviors stem from the policy optimization process or the reward learning process. The latter approach remains the question of whether such evaluation accurately predicts the performance of the optimized policy.

---

*This work was done when Xueru Wen interned at Xiaohongshu.
†Corresponding author.

In this work, we systematically investigate the effectiveness of using accuracy in predicting downstream performance. Essentially, the accuracy metric measures the difference between two reward functions, referred to as **RM error**. In practice, the accuracy quantifies the error between the learned reward model and the empirical human rewards. The RM error can lead to potential performance loss between the policies optimized for the proxy RM and the golden reward function. Specifically, we define this performance gap as **Policy Regret**. Therefore, the goal of RM evaluation is to predict policy regret through RM error, as depicted in Figure 1. To investigate how differences in RM measured by accuracy translate into gaps in optimized policy performance, we design a synthetic experiment framework. Due to the inaccessibility of human reward functions, we employ synthetic RM as the golden reward function in our experiments. To effectively collect golden-proxy RM pairs for analysis, we create $N$ different RMs, designating one RM as the golden model and the others as proxy models each time. Based on the above framework, we decompose the issue concerning RM evaluation into three research questions (RQ):

**RQ1: Does the RM error measured by accuracy correlate with policy regret?** We examine the correlation between accuracy and policy regret on the widely adopted RewardBench dataset(Lambert et al., 2024). We employ optimization algorithms, i.e., best-of-$n$ sampling and policy gradient-based reinforcement learning, for investigation. Our findings reveal a weak positive correlation between the measured accuracy and the policy regret. However, we observe that policies optimized towards reward models within a similar accuracy range can have quite different regrets.

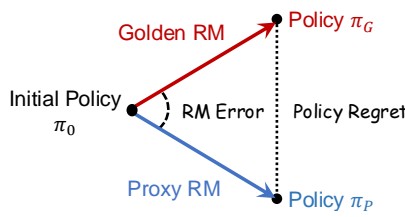

Figure 1: Translation from RM error to policy regret.

**RQ2: How to better measure RM error for policy regret prediction?** While we present a positive correlation between the accuracy and the policy regret, there remains room for further enhancement. This naturally raises the question of how to better quantify the RM error to achieve a stronger correlation. We begin by investigating the influence of prompt and response distributions. Specifically, we observe that the prompt difference between the RM test dataset and the downstream test dataset will potentially decrease the correlation between the accuracy and the policy regret. Regarding response distribution, we find that the rank of responses evaluated by the golden RM impacts correlation more significantly than the model from which they are sampled. Furthermore, we propose a simple yet effective strategy to enhance correlation: increasing the number of responses per prompt in the test dataset. We validate this approach under constraints of equal sample size and annotation budget, finding that metrics based on more responses achieve a higher correlation with policy regret.

**RQ3: What's the relationship between RM error and Policy Regret?** The translation from RM error to policy regret stems from the overoptimization of the reward model. Specifically, optimizing toward an imperfect proxy RM leads to a deterioration in downstream performance. Therefore, to predict the policy regret, the quantified RM error should be able to reflect the potential overoptimization. In this work, we derive the expected relationship between the accuracy and the degree of reward overoptimization under the assumptions of Regressional Goodhart's effect (Manheim & Garrabrant, 2019) and normal distributions of both reward score and noise. However, our findings indicate that RMs with similar accuracy can behave quite differently in terms of overoptimization. This indicates accuracy alone can be an inadequate metric for predicting the downstream performance.

In summary, our work provides deeper understanding into the relationship between RM error and policy regret. This offers valuable insights for both reward model training and the development of a more robust RLHF algorithm. Our results also highlight the need for a more rigorous benchmark in the evaluation of reward models and the development of advanced RM assessment tools.

## 2 PRELIMINARY

Let $\mathcal{X}$ be the set of prompts and $\mathcal{Y}_{\mathcal{X}}$ be the set of possible responses. A policy $\pi$ is a language model that generates responses $y$ to the prompt $x$ at probability $\pi(y|x)$. The golden reward function assesses the response and gives a score based on their quality $r^* : \mathcal{X} \times \mathcal{Y}_{\mathcal{X}} \to \mathbb{R}$. In practice, the golden reward function $r$ represents complicated human preference and is generally inaccessible. Instead, a reward model $r$ learned from preference data serves as a proxy for the golden reward

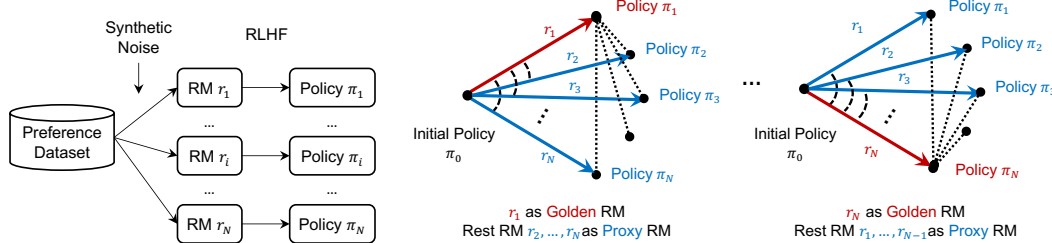

(a) Synthetic pipeline of Reinforcement Learning from Human Feedback.

(b) Proxy-golden RM pairs collection for evaluating the correlation between the RM error and policy regret.

Figure 2: Overall experiment framework.

function $r^*$. Then policy $\pi$ is optimized to maximize the expectation of the reward given by $r$:

$$\pi = \max_{\pi} \mathbb{E}_{x \in \mathcal{X}, y \sim \pi(\cdot|x)}[r(x, y)] \tag{1}$$

This results in a sub-optimal policy $\pi$ rather than the ideal policy $\pi^*$ that directly maximizes the reward given by the golden reward function $r^*$.

The workflow of Reinforcement Learning from Human Feedback (RLHF) can be represented as:

- $\mathcal{A}_{RM}(D_{RM}^{train}) \to r$: Learn a proxy RM $r$ on the preference dataset $D_{RM}^{train}$ by algorithm $\mathcal{A}_{RM}$.

- $\mathcal{M}(r, r^*|D_{RM}^{test}) \to d_r$: Evaluate the RM error between proxy reward model $r$ and golden reward function $r^*$ by metric $\mathcal{M}$ on the test dataset $D_{RM}^{test}$.

- $\mathcal{A}_{RL}(\pi_0, r|D_{RL}^{train}) \to \pi$: Optimize the initial policy $\pi_0$ and get policy $\pi$ that maximizes the expectation of reward given by the proxy RM $r$ by the algorithm $\mathcal{A}_{RL}$.

- $\mathcal{M}'(\pi, \pi^*|D_{RL}^{test}) \to d_\pi$: Measure the regret between the policy $\pi$ and the policy $\pi^*$ that optimized directly towards $r^*$ by metric $\mathcal{M}'$ on the test dataset $D_{RL}^{test}$.

In this work, we essentially focus on the relationship between the error of reward model $d_r$ and the regret of policy $d_\pi$, which translates our issue into: **How should we measure the error $d_r$ so as to reach high correlation with the policy regret $d_\pi$?** Various factors introduced in the RLHF process can potentially influence this relationship. To systematically investigate their impact, we focus on several key factors while keeping other variables fixed. The test dataset $D_{RM}^{test}$ on which the reward model $r$ is evaluated; The metric $\mathcal{M}$ used to evaluate the error between the reward models; The test dataset $D_{RL}^{test}$ on which the final policy $\pi$ is evaluated.

One challenge in conducting the investigation is the inaccessibility of the golden reward function representing human preferences. To address this difficulty, we follow the previous practice (Gao et al., 2022; 2024) of utilizing another reward model to act as the golden reward function.

Another challenge is to effectively collect the proxy-golden RM pairs for evaluating the correlation between the RM error and policy regret. We implement this by constructing $N = 10$ different reward models $r_i$ by randomly flipping $\alpha_i$ of the pairs in the training dataset[1] $D_{RM}^{train}$. Then $N$ policies $\pi_i$ will be optimized towards the corresponding reward model $r_i$. This pipeline is demonstrated in Figure 2a. This operation will form $N \times (N-1)$ golden-proxy RM pairs. As shown in Figure 2b, these pair collections can then be used for computing correlation by letting one reward model as the golden reward function and the remaining reward models as proxy reward models.

Building on prior works that formally define policy regret in the context of general Markov Decision Processes (MDPs) (Karwowski et al., 2023; Fluri et al., 2024), we adapt the concept within the framework of RLHF. Given that the KL divergence between a policy $\pi$ and the initial policy $\pi_0$ is

---

[1] We constructed the data such that if a training set has $\alpha_i$ of its labels flipped and another has $\alpha_j$ flipped, then $|\alpha_i - \alpha_j|\%$ of the labels are inconsistent between the two sets. See Appendix 8.1 for dataset details.

$KL(\pi||\pi_0) = \lambda$, the regret with respect to a golden reward function $r^*$ is defined as:

$$Reg_{r^*} = \frac{\max\limits_{KL(\pi'||\pi_0)<\lambda} J_{r^*}(\pi') - J_{r^*}(\pi)}{\max\limits_{KL(\pi'||\pi_0)<\lambda} J_{r^*}(\pi') - \min\limits_{KL(\pi'||\pi_0)<\lambda} J_{r^*}(\pi')} \quad (2)$$

Here, the function $J_{r^*}(\pi) = \mathbb{E}_{y \sim \pi(\cdot|x)}[r^*(x,y)]$ represents the expected reward given the policy $\pi$ and the golden reward function $r^*$. This value reflects the ratio of the maximum reward gain a policy can achieve to the actual reward gain obtained, considering the constraint of KL divergence.

However, this definition is hard to compute due to the inaccessibility of the optimal policy and the difficulty of controlling KL divergence $\lambda$. Therefore, we propose **Normalised Drop Ratio (NDR)** as a computable estimation as follows:

$$\mathcal{M}'_{NDR}(\pi, \pi^*) = \frac{J_{r^*}(\pi) - J_{r^*}(\pi_0)}{J_{r^*}(\pi^*) - J_{r^*}(\pi_0)} \quad (3)$$

where the policies $\pi$ and $\pi^*$ are achieved by optimizing initial policy $\pi_0$ against the golden reward function $r^*$ and the proxy RM $r$ with exactly the same hyperparameters. This metric quantifies the performance deterioration resulting from optimizing against a proxy RM $r$ instead of the golden reward function $r^*$. It also accounts for variations due to different scales among different RMs.

The data from the RewardBench (Lambert et al., 2024) are used to construct the test datasets $D_{RM}^{test}$ and $D_{RL}^{test}$. All RMs are initialized from the Llama-3-instruct-8B model (AI@Meta, 2024) and finetuned by minimizing the negative log-likelihood loss with regularization term (Hou et al., 2024):

$$\mathcal{L}_{RM} = -\mathbb{E}_{(x,y_w,y_l) \sim D_{RM}^{train}}[\log(\sigma(r(x,y_w) - r(x,y_l)))] - \mathbb{E}_{(x,y) \sim D_{RM}^{train}}[r(x,y)^2] \quad (4)$$

We adopt best-of-$n$ sampling (BoN) and PPO (Schulman et al., 2017) as the algorithm $A_{RL}$ to optimize the initial policy $\pi_0$, which is also the Llama-3-instruct-8B model. For the best-of-$n$ sampling, the reward model $r$ picks the response with the highest reward from $n$ candidates. For the PPO algorithm, we follow (Gao et al., 2022) and set the KL penalty in all experiments to be 0.

## 3 DOES THE RM ERROR MEASURED BY ACCURACY CORRELATE WITH THE POLICY REGRET?

Current research typically assesses reward model errors by computing accuracy on a fixed test set. Although widely used, it remains unclear whether the accuracy correlates with policy regret. In this section, we evaluate the accuracy and policy regret using the RewardBench (Lambert et al., 2024) dataset. We perform deduplication on the prompts, leaving 2,733 distinct prompts.

**Finding 1:** *RM evaluation accuracy is positively related to policy regret, but even with similar RM accuracy, policies can exhibit different levels of regret.*

We first investigate the correlation between accuracy and policy regret. Original RewardBench prompts and responses are used to measure accuracy, and the prompts are used for downstream optimization. We calculate the correlations between RM error, measured by accuracy, and policy regret, measured by NDR, in Table 1. As presented in Figure 3a and Figure 3b, there is a positive relationship between accuracy and policy regret. However, trends illustrate that policy regret can vary considerably even within similar accuracy ranges. Lastly, we observe that accuracy generally correlates more strongly with regret in BoN than in PPO. This is expected, as BoN is a more localized and stable optimization algorithm, making it more predictable by reward model error.

## 4 HOW TO BETTER MEASURE RM ERROR FOR POLICY REGRET PREDICTION?

In the previous section, we examine the positive correlation between accuracy and policy regret. However, there appears to be room for enhancing this correlation, which leads us to the question: how to better quantify the RM error? In this section, we first investigate the influence of prompt and response distribution. Moreover, we explore a straightforward yet effective strategy, i.e., extending responses per prompt. Finally, we validate it under different constraints.

Table 1: The correlation between the accuracy (**Acc.**) and the policy regret under BoN and PPO measured by typical correlation coefficients and Mean Reciprocal Rank (**MRR**). MRR is calculated by averaging the reciprocal ranks of policies optimized by the RM with the highest accuracy.

| Error / Regret | | Relevance | | | |
|---|---|---|---|---|---|
| | | Kendall $\tau$ corr. | Pearson corr. | Spearman corr. | MRR |
| Acc. | BoN | 0.6561 | 0.7520 | 0.7533 | 0.6333 |
| Acc. | PPO | 0.4654 | 0.6395 | 0.6102 | 0.5167 |

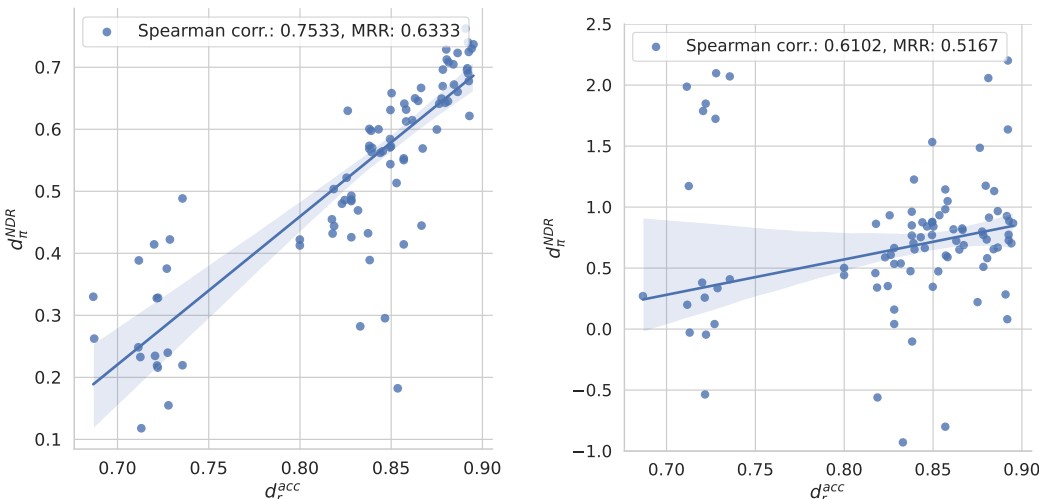

(a) Trend between accuracy $d_r^{acc}$ and the policy regret $d_\pi^{NDR}$ under BoN setting.

(b) Trend between the accuracy $d_r^{acc}$ and the policy regret $d_\pi^{NDR}$ under PPO setting.

Figure 3: Trend between the accuracy and policy regret measured by the normalised drop ratio.

**Finding 2:** *The rank of responses affects the correlation between accuracy and regret more than the response sampling models.*

In the previous experiment, we used the original responses in RewardBench for validation, which were sampled from different models. This naturally raises the question: **Is it possible to achieve a higher correlation by sampling responses exclusively from the model used for downstream optimization?** To examine this question, we construct multiple test datasets, each containing responses exclusively sampled from a single downstream model.. We then evaluate the correlation between policy regret and computed accuracy. As shown in Table 2, this approach does not consistently improve upon the original RewardBench dataset. This result suggests that sampling responses from the model used for optimization may not be necessary to achieve a strong correlation.

While it seems unnecessary to sample from models used for optimization, we observe varying correlations in datasets constructed with responses by different models. This indicates that other factors, such as **the rank of chosen and rejected samples**, may be at play. To examine this assumption, we prepare multiple responses for all prompts. Each time, we sort the responses by golden rewards given by one RM, and divide them into bins. We then randomly sample accepted and rejected responses from different bins to construct test datasets with responses of varying ranks. The accuracy on these test datasets is used to assess the correlation with policy regrets. As demonstrated in Figure 4a and Figure 4b, different trends emerged for BoN and PPO. For BoN, test datasets with chosen samples from mid-rank bins and rejected samples from lower-rank bins exhibited higher correlations. In contrast, for PPO, datasets with chosen samples from higher-rank bins and rejected samples from mid-rank bins showed a stronger correlation. This distinction likely stems from the characteristics of the two optimization algorithms. With PPO, the policy is optimized more aggressively, causing

Table 2: The correlation between policy regret and accuracy on datasets with responses sampled from different models. The **Origin** represents the result on the RewardBench dataset. The highest results are highlighted in **bold**. The model used for downstream optimization is suffixed with *.

| Model | BoN | | PPO | |
|---|---|---|---|---|
| | Spearman corr. | MRR | Spearman corr. | MRR |
| Mistral-7B | 0.680±0.042 | 0.579±0.079 | 0.580±0.037 | 0.573±0.073 |
| Llama3-8B* | 0.644±0.038 | 0.598±0.080 | **0.648±0.047** | 0.551±0.061 |
| Qwen2-7B | 0.680±0.030 | 0.529±0.074 | 0.603±0.039 | **0.599±0.060** |
| Vicuna-7b | 0.703±0.035 | **0.658±0.075** | 0.527±0.036 | 0.517±0.068 |
| Origin | **0.753** | 0.633 | 0.610 | 0.517 |

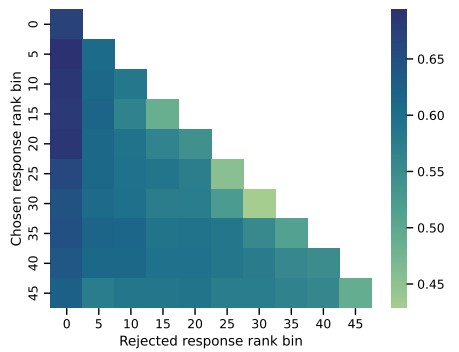

(a) The correlation trend on BoN.

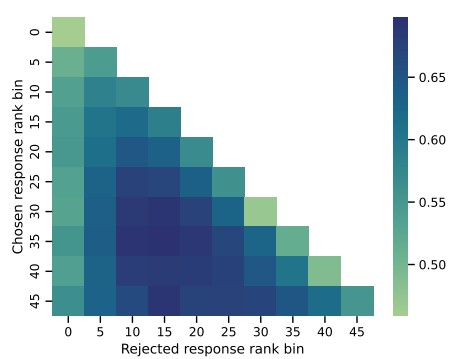

(b) The correlation trend on PPO.

Figure 4: The correlation between policy regret and accuracy on datasets constructed from responses. Each grid on the x and y axes represents a sampled bin, with the values below indicating the corresponding rank. For example, the 5 on the y-axis indicates that the chosen responses are sampled from those ranked between 5 and 10.

responses to rank higher under the golden RM, making the consistency between the proxy RM and the golden RM on high-reward samples more critical. Conversely, for BoN, where the degree of policy optimization is less intense, consistency on mid-reward samples can be more instructive.

**Finding 3:** *The prompt difference between the RM test dataset and downstream test dataset can weaken the accuracy-regret correlation.*

The previous experiments assume that the RM and RL test datasets consisted of the same prompts, which may not be true in practice. To investigate **the influence of prompt category**, wWe split the test dataset based on the prompt categories determined by the original classifications in Reward-Bench. We evaluate accuracy and regret in each category and demonstrate the results in Table 3. We find that accuracies in each category align more closely with regret in the corresponding category under the BoN setting. However, this relationship is not present in the PPO setting. This may be because different types of prompts influence each other during PPO optimization.

To further investigate **the influence of prompt semantics**, we randomly rewrite some prompts in the original test dataset, creating new test datasets with prompts different from the downstream dataset. We ask GPT-4o (OpenAI, 2024) to perform two different strategies on these prompts, as detailed in Appendix 8.2. One strategy alters the expression without changing the meaning, and the other generates a new prompt within the same category. As shown in Figure 5, we find that the correlation is less affected by paraphrasing under BoN. However, the correlation measured by the Spearman coefficient on PPO continuously weakens as we paraphrase a larger ratio of prompts.

Table 3: The correlation between policy regret and accuracy on the test dataset composed of prompts from different categories evaluated by the Spearman coefficient. The highest result in each column is **bolded**, and the highest in each row is underlined.

| Regret | | Accuracy | | | | |
|---|---|---|---|---|---|---|
| | | Chat | ChatHard | Code | Math | Safety |
| BoN | Chat | **0.529±0.082** | 0.682±0.058 | 0.573±0.050 | 0.557±0.052 | 0.657±0.044 |
| | ChatHard | 0.493±0.089 | 0.682±0.053 | 0.583±0.038 | 0.538±0.066 | 0.655±0.051 |
| | Code | 0.504±0.095 | 0.634±0.059 | **0.717±0.043** | 0.456±0.060 | 0.646±0.053 |
| | Math | 0.288±0.121 | 0.343±0.080 | 0.244±0.048 | **0.610±0.058** | 0.282±0.059 |
| | Safety | 0.515±0.093 | **0.705±0.057** | 0.521±0.047 | 0.497±0.067 | **0.687±0.049** |
| PPO | Chat | 0.349±0.105 | 0.500±0.086 | 0.441±0.054 | 0.192±0.083 | **0.611±0.061** |
| | ChatHard | 0.314±0.115 | 0.484±0.083 | 0.434±0.057 | 0.202±0.087 | 0.576±0.073 |
| | Code | 0.384±0.092 | 0.450±0.080 | **0.527±0.051** | **0.371±0.060** | 0.459±0.053 |
| | Math | 0.185±0.091 | 0.275±0.064 | 0.238±0.050 | 0.145±0.057 | 0.313±0.054 |
| | Safety | **0.359±0.118** | **0.521±0.062** | 0.442±0.061 | 0.332±0.069 | 0.533±0.053 |

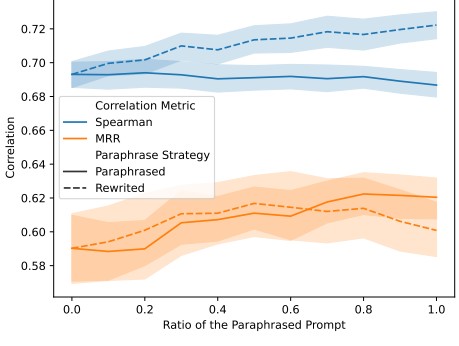

(a) The correlation trend on BoN.

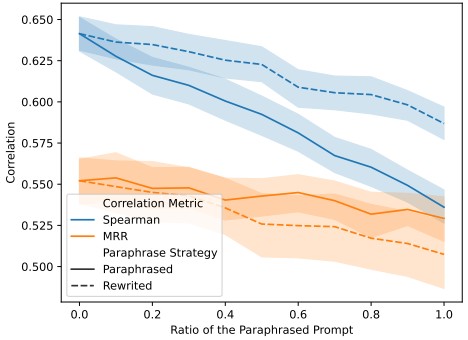

(b) The correlation trend on PPO.

Figure 5: The correlation between policy regret and accuracy on datasets with different ratios of paraphrased prompts evaluated by the Spearman coefficient and MRR. Different line styles are used to represent datasets paraphrased by different strategies.

**Finding 4:** *Increasing the number of responses per prompt can enhance the correlation between measured RM error and policy regret.*

We first examine the strategy of response expansion per prompt by preparing 5 responses per prompt and **employing various metrics**. The details about the computation of these metrics can be found in Appendix 8.8. These metrics include common correlation measures, as well as metrics adapted from Information Retrieval. We also explored other metrics for comparison, e.g., expected calibration error and Bo5. Note that some metrics require absolute scoring of responses, which may be inaccessible in practice. As shown in Table 4, metrics evaluated on the dataset with more responses consistently achieve higher correlations in comparison to the accuracy on the dataset with two responses per prompt. Moreover, the $\xi$ correlation (Chatterjee, 2020) generally demonstrated the best performance. The ECE metric, despite its relative commonness, does not show a significant correlation with regret.

Although measuring more responses per prompt yields a higher correlation, it can be unfair without controlling for the total number of samples (prompt-response pairs) in the test dataset. Thus, we further explore the question of whether we should increase the number of responses or prompts while maintaining a constant sample size. We fix the sample size in the test dataset while varying

Table 4: The correlation between different metrics and policy regret. The **Accuracy-pair** stands for the baseline that only samples two responses per prompt and computes the accuracy. Metrics that require an absolute score for each response are suffixed with *.

| Metrics | BoN | | PPO | |
|---|---|---|---|---|
| | Spearman corr. | MRR | Spearman corr. | MRR |
| Pearson corr.* | 0.663±0.012 | 0.657±0.040 | 0.685±0.016 | 0.545±0.020 |
| Spearman corr. | 0.664±0.015 | 0.664±0.046 | 0.680±0.019 | 0.552±0.030 |
| Kendall $\tau$ corr. | 0.665±0.016 | **0.667±0.042** | 0.686±0.019 | 0.558±0.035 |
| Accuracy | 0.656±0.015 | 0.648±0.036 | 0.671±0.018 | 0.554±0.025 |
| Bo5* | 0.666±0.020 | 0.663±0.058 | 0.670±0.024 | 0.543±0.032 |
| ECE | 0.173±0.016 | 0.364±0.027 | 0.063±0.014 | 0.382±0.022 |
| MRR | 0.650±0.018 | 0.646±0.054 | 0.675±0.022 | 0.565±0.042 |
| NDCG | 0.655±0.032 | 0.614±0.071 | 0.658±0.039 | 0.562±0.053 |
| $\xi$ corr. | **0.677±0.026** | 0.649±0.051 | **0.688±0.026** | **0.583±0.047** |
| Accuracy-pair | 0.635±0.039 | 0.586±0.080 | 0.642±0.047 | 0.567±0.082 |

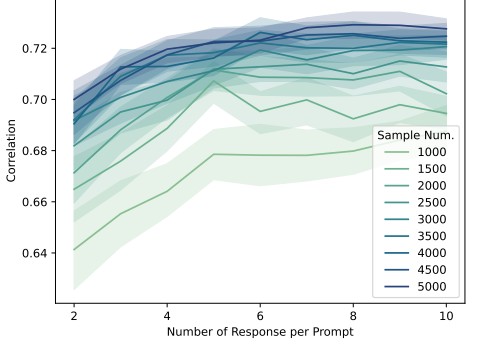

(a) The correlation trend on BoN.

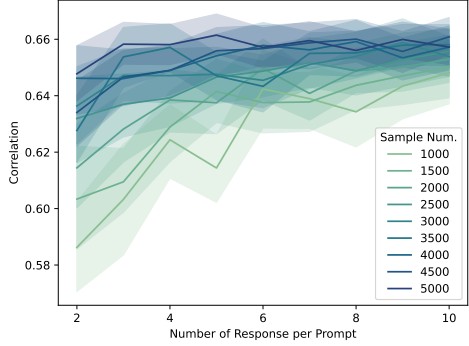

(b) The correlation trend on PPO.

Figure 6: The Spearman coefficient between accuracy and policy regret with test datasets having the same number of samples but varying numbers of responses per prompt.

the number of responses. As shown in Figure 6a and Figure 6b, it is generally more effective to increase the number of responses rather than the number of prompts. However, this advantage decreases with the growing sample size, possibly due to reaching an upper bound in the correlation between regret and measured accuracy.

The annotation cost can be another important factor since building a test dataset with more than two responses can be tricky. To explore this factor, we further examine the question that **should we increase responses or prompts under a constant annotation budget**. To conduct the experiment, we consider the noise in the annotation process. Among various probability ranking models (Critchlow et al., 1991), we adopt the most commonly used one, i.e., the Bradley-Terry model (Bradley & Terry, 1952). Given a prompt $x$ and two responses $y_i$ and $y_j$, the probability that $y_i$ is preferred to $y_j$ is computed by:

$$P(y_i \succ y_j | x) = \frac{1}{1 + e^{(r_j - r_i)/\beta}} \tag{5}$$

where $r_i$ and $r_j$ is the golden reward score given to the response $y_i$ and $y_j$ in our case. The parameter $\beta$, which can be seen as an indicator of the annotation quality, is set to $0.5$. We evaluate the annotation cost by measuring the number of pairwise comparisons. This setting fits the Bradley-Terry model and is also commonly adopted in real-world practice. The annotation process is stimulated by performing a sorting algorithm based on pair-wise comparison. The final results are presented in

Figures 7a and 7b. We find that under the BoN setting, including more responses is more beneficial with the same annotation budget. However, with a limited annotation budget, the benefits quickly diminish as the number of responses grows. Additionally, in the PPO setting, increasing the number of responses does not yield significant advantages.

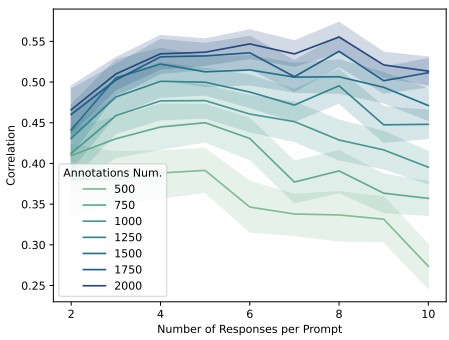
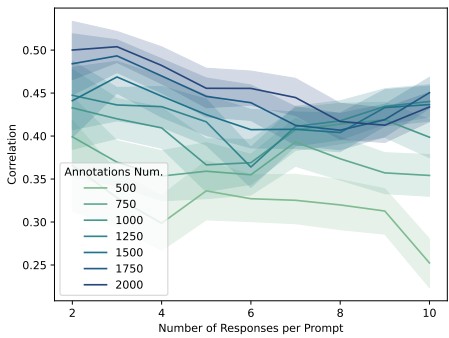

(a) The correlation trend on BoN.  (b) The correlation trend on PPO.

Figure 7: The Spearman coefficient between policy regret and accuracy on test datasets with the same annotation budget but varying the number of responses per prompt.

# 5 WHAT'S THE RELATIONSHIP BETWEEN RM ERROR AND POLICY REGRET?

In this section, we investigate the relationship between RM error and policy regret. The translation from RM error to the policy regret can be seen as the result of the reward model overoptimization. Therefore, to accurately predict policy regret, the quantified RM error should be able to reveal the potential overoptimization. We first theoretically explore this issue. Then we empirically examine the relationship and analyze the observed optimization dynamics.

**Finding 5:** *Accuracy alone can be insufficient to capture the potential RM overoptimization.*

Goodhart's Law (Goodhart, 1984) is the main cause for the translation from RM error to policy regret. It states that optimizing a less effective metric rather than the golden metric leads to system failure. Such a phenomenon is also commonly termed reward model overoptimization (Gao et al., 2022) under the context of RLHF. Among the various kinds of Goodhart's effect that lead to reward model overoptimization, the Regressional Goodhart effect (Manheim & Garrabrant, 2019) is the most fundamental and unavoidable. It occurs when the proxy reward model $r$ is essentially the golden reward function $r^*$ mixed with some independent noise $z$, namely:

$$r = r^* + z \tag{6}$$

Assuming that only the Regressional Goodhart effect occurs and both the reward score and noise are normally distributed, i.e. $r^* \sim \mathcal{N}(0, \sigma_r^2)$ and $z \sim \mathcal{N}(0, \sigma^2)$, the relationship between accuracy $d_r^{acc}$ and the degree of overoptimization $d_\pi$ [2] can be expressed by the following parametric equations:

$$
\begin{aligned}
d_r^{acc} &= 1 - \int_{x=0}^{+\infty} \frac{1}{\sigma_r \sqrt{\pi}} e^{-\frac{x^2}{4\sigma_r^2}} \Phi\left(-\frac{x}{\sqrt{2}\sigma}\right) dx \\
d_\pi &= \frac{\sigma_r^2}{\sigma_r^2 + \sigma^2}
\end{aligned}
\tag{7}
$$

where $\Phi$ denotes the cumulative distribution function of the standard normal distribution.. The equation is too complex to be solved analytically, so we resort to a numerical solution. We plot the expected relationship between accuracy and the degree of overoptimization in Figure 8a.

---

[2]The definition of $d_\pi$ and detailed derivation are provided in Appendix 8.9.

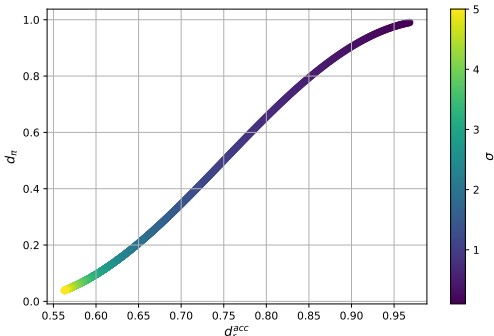 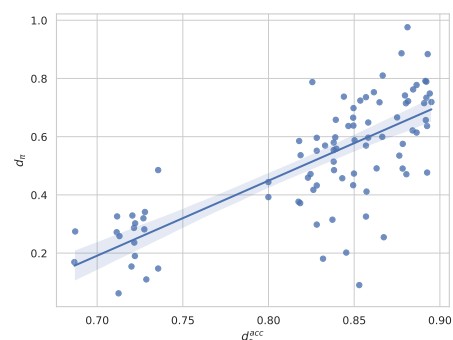

(a) Expected trend of accuracy $d_r^{acc}$ and the degree of overoptimization $d_\pi$ by varying the noise $\sigma$.

(b) The trend of the accuracy $d_r^{acc}$ and the estimated degree of overoptimization $d_\pi$ under BoN.

Figure 8: Trends between the accuracy $d_r^{acc}$ and the degree of overoptimization $d_\pi$.

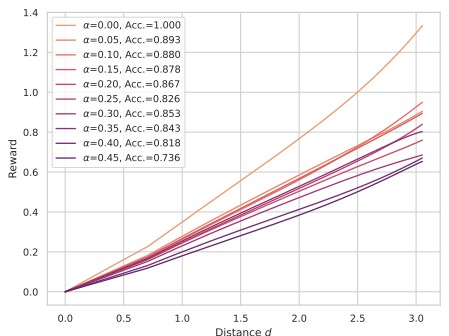 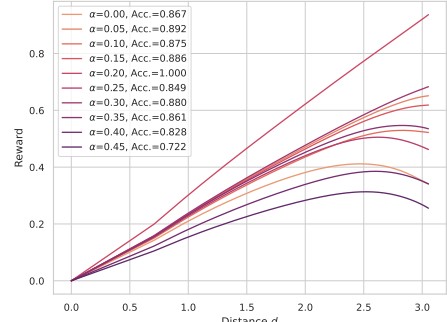

(a) RM with $\alpha = 0$ as golden reward model.

(b) RM with $\alpha = 0.15$ as golden reward model.

Figure 9: Overoptimization trends under BoN setting. The change of golden reward score when optimizing towards different proxy RMs are plotted. The distance $d$ is computed as $\sqrt{D_{KL}(\pi||\pi_0)}$ (Bai et al., 2022), where the KL divergence of best-of-$n$ sampling can be computed by $\log n - \frac{n-1}{n}$ (Hilton, 2023). The rewards are computed by an unbiased estimator (Nakano et al., 2022).

However, the trend under the BoN setting, as plotted in Figure 8b, reveals outliers beyond the expected relationship. These discrepancies may stem from other forms of Goodhart's effects (Manheim & Garrabrant, 2019) beyond the Regressional type. Figure 9 illustrates the optimization dynamics when using different golden RMs and optimizing towards various proxy RMs. We observe distinct overoptimization phenomena for golden-proxy RM pairs that have similar accuracy levels. In Figure 9a, there is nearly no drop in golden reward scores as the KL divergence increases, indicating that the Regressional Goodhart's effect is dominant in this case. In contrast, Figure 9b shows a noticeable decline in golden reward scores across nearly all proxy RMs, suggesting the influence of more complex Goodhart's effects. These differences suggest that solely relying on accuracy may be insufficient to predict potential overoptimization.

## 6 RELATED WORKS

Reinforcement Learning from Human Feedback (RLHF) (Bai et al., 2022; Ouyang et al., 2022; Stiennon et al., 2022) has been a common strategy for the alignment of Large Language Models (Yang et al., 2024; Dubey et al., 2024), in which the reward model plays a crucial role. However, this methodology faces several challenges (Casper et al., 2023; Lang et al., 2024; Armstrong & Mindermann, 2019; Skalse et al., 2022), such as reward model overoptimization (Gao et al., 2022;

Lehman et al., 2019), where optimizing policy towards proxy reward model may lead to the ground truth performance deterioration. This phenomenon, as a special case of reward hacking (Krakovna, 2020), can seen as the result of Goodhart's law (Goodhart, 1984; Zhuang & Hadfield-Menell, 2021).

The RM as an imperfect proxy of the golden preference necessitates the evaluation of whether the reward model accurately captures the real preference (Michaud et al., 2020; Russell & Santos, 2019). Many works consider the measurements to predict the potential consequence due to the difference of reward functions (Ng et al., 1999; Skalse et al., 2023). (Gleave et al., 2021; Skalse et al., 2024) propose the metric quantify the difference between the reward functions and induce regret bounds for optimal policies. However, some works (Zhuang & Hadfield-Menell, 2021; Fluri et al., 2024) propose negative visions of optimization towards imperfect proxy RMs. These works assume the accessibility of the golden reward function, which can be false under the RLHF setting. The common practice under RLHF is to compute accuracy on a fixed preference dataset (Lambert et al., 2024), which remains the question about theoretical or empirical validation of performance prediction (Zhou et al., 2024; Kim et al., 2024).

## 7 CONCLUSION

Our study highlights that although there is a weak positive correlation between accuracy and policy performance, RMs with similar accuracy can result in varying policy outcomes. Moreover, the method used to measure accuracy significantly impacts prediction performance. Finally, we find that accuracy alone may not fully reflect the phenomenon of overoptimization. Overall, we suggest a more cautious attitude about RM performance as indicated by accuracy and emphasize the importance of developing more sophisticated RM evaluation techniques..

ACKNOWLEDGMENTS

We sincerely thank the reviewers for their insightful comments and valuable suggestions. This work was supported by Beijing Natural Science Foundation (L243006), CAS Project for Young Scientists in Basic Research (Grant No.YSBR-040), the Natural Science Foundation of China (No. 62306303, 62272439).

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

## 8 APPENDIX

### 8.1 DATASET CONSTRUCTION

We build up a RM training dataset by mixing the following open-sourced datasets:

- **Nectar** (Zhu et al., 2023), a high-quality 7-wise comparison dataset generated by GPT-4 ranking.
- **Capybara-7K-binarized** (Argilla, 2024), a binarized prefernce dataset built with distilabel atop the **Capybara** (Daniele & Suphavadeeprasit, 2023).
- **Orca-pairs** (Intel, 2023), a dataset contains 12k examples from OpenOrca dataset (Lian et al., 2023).
- **UltraFeedback** (Cui et al., 2023), a large-scale, fine-grained, diverse preference dataset.
- **PKU-SafeRLHF** (Ji et al., 2024), a high-quality dataset consisting of 83.4K preference entries, which is annotated across two dimensions: harmlessness and helpfulness.
- **MTBench-human** (Zheng et al., 2023), a dataset contains 3.3K expert-level pairwise human preferences for model responses generated by 6 models in response to 80 MT-bench questions.
- **Chatbot-arena** (Zheng et al., 2023), a dataset contains 33K cleaned conversations with pairwise human preferences. Responses generated by 6 models in response to 80 MT-bench questions.
- **HH-RLHF** (Bai et al., 2022; Ganguli et al., 2022), a dataset contains human preference data about helpfulness and harmlessness.

We retain only single-turn dialogue data, deduplicating based on prompt string matching. Next, we filtered out excessively long samples and balanced the proportion of positive and negative sample lengths. Ultimately, we retained 112400 preference data samples, with 7052 set aside as a validation set for reward model training.

### 8.2 PARAPHRASE PROMPTS

We employed two strategies for rewriting prompts: one that alters expression without changing semantics, and another that rewrites the prompt into a similar but related prompt. The prompt in Table 5 is used for the paraphrase strategy, and the prompt in Table 6 is used for the rewrite strategy.

### 8.3 RELATIONSHIP BETWEEN GOLDEN-PROXY PAIRS

In this section, we present the relationship between golden-proxy pairs from our experiments. First, we illustrate the accuracy between golden RMs and proxy RMs in Figure 10. As shown, the image is symmetrical, and generally, as the data noise decreases, the accuracy between each golden-proxy pair increases. Next, we display the relationship of the NDR metric between proxy-golden pairs under BoN and PPO optimization. These figures do not exhibit obvious symmetry, and there are more extreme values under PPO optimization.

Table 5: The prompt used for paraphrase strategy.

Now you should play the role of prompt engineer.
Your task is to paraphrase the prompt into a new prompt.
If the origin prompt is a safety-related prompt, paraphrase it into a new safe prompt.
When outputting, only provide the paraphrased prompt and nothing else.

>> Input Prompt:
{prompt}

>> Output:

Table 6: The prompt used for rewrite strategy.

Now you should play the role of prompt engineer.
You should generate a new prompt of the same category with an input prompt.
If the origin prompt is a safety-related prompt, paraphrase it into a new safe prompt.
When outputting, only provide the generated prompt and nothing else.

>> Input Prompt:
{prompt}

>> Output:

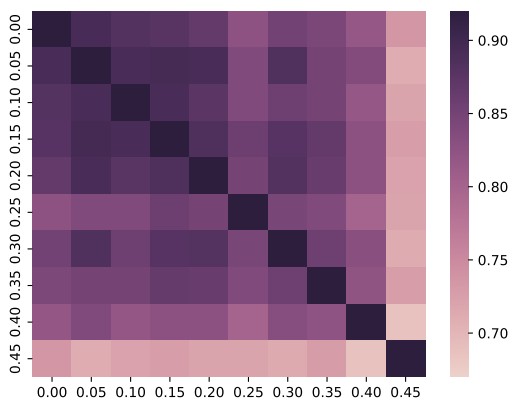

Figure 10: The accuracy between golden and proxy RMs across varying noise levels in training data.

## 8.4 CORRELATION BETWEEN ACCURACY AND DOWNSTREAM PERFORMANCE

In real-world scenarios, various factors can lead to discrepancies between proxy RMs and golden RMs. In the main experiments, we primarily considered noise in the training dataset. In this section, we explore two other common scenarios: 1) differences in model architecture between the proxy RM and the golden RM, and 2) differences in training data. For the first scenario, we train RMs using various Qwen2.5 Team (2024) series pretrain and instruct models, ranging from 0.5 to 72 billion parameters. For the second scenario, we randomly divide the training into $N = 10$ parts, using the first $i$ parts to train the $i$-th RM. We then measure the correlation between the accuracy of these two types of RMs and the policy regret optimized through BoN, as shown in Figure 12a and Figure 12b. The results indicate that the correlation for RMs trained on different datasets is significantly lower than for those trained on the same dataset with different models, possibly due to their high similarity indicated by accuracy.

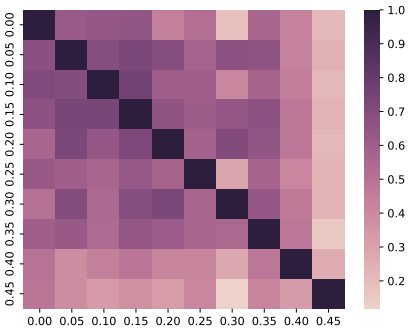 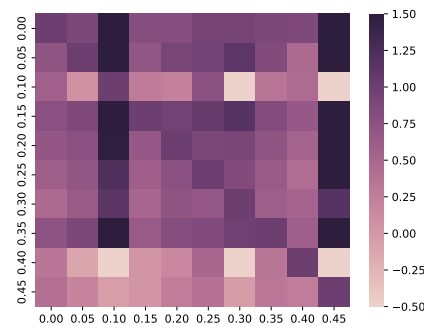

(a) The NDR metric under BoN setting.   (b) The NDR metric under PPO setting.

Figure 11: The NDR metrics between proxy-golden pairs under different settings.

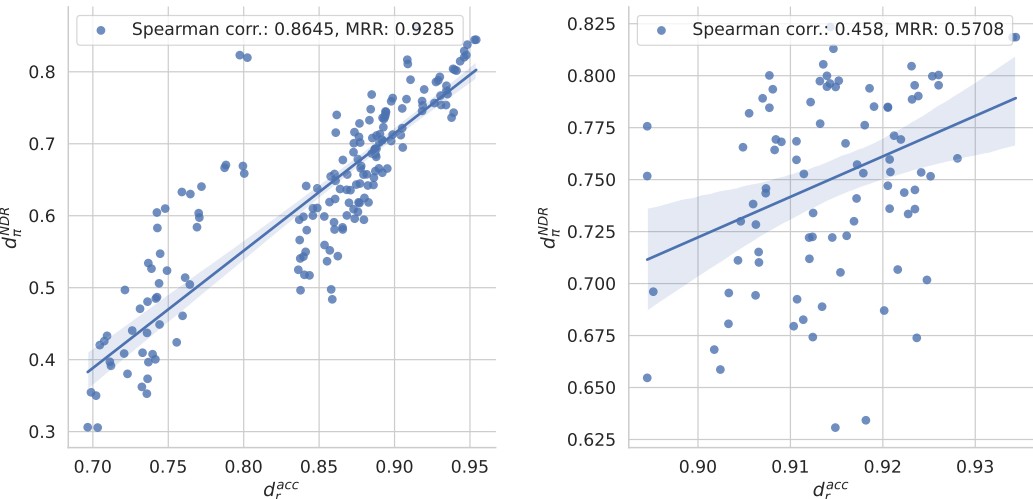

(a) The correlation trend under model scaling setting.   (b) The correlation trend under dataset split setting.

Figure 12: The correlation between accuracy and policy regret for RMs of different settings.

Beyond synthetic environments, we also explore the relationship between RM metrics and downstream task performance in more realistic scenarios. We assess the accuracy of various open-source RMs on RewardBench Lambert et al. (2024) and examine their correlation with downstream tasks using MT-Bench Zheng et al. (2023), evaluated with GPT-4o. For evaluating downstream task performance, we optimize LLaMA-3-Instruct (AI@Meta, 2024) using Best-of-32. The results, shown in Figure 13, indicate a positive correlation between RewardBench metrics and MT-Bench scores. However, RMs with similar accuracy can lead to fluctuating downstream performance.

## 8.5 INFLUENCE OF KL PENALTY

In the main experiment, the KL penalty was set to $0$. However, there remains the question of whether a larger KL penalty might increase the correlation with PPO. In this section, we examine the correlation between accuracy and PPO NDR using the same RMs from the main experiment but with varying KL penalties. The results in Table 7 indicate that appropriately increasing the KL penalty enhance the Spearman correlation while reduces the MRR. When the KL penalty becomes too large, the Spearman correlation also decreases. We believe that with a smaller KL penalty, the PPO optimization process becomes more localized and stable, thereby strengthening the predictive correlation of RM error. However, the presence of the KL penalty limits the consistent increase of the expected

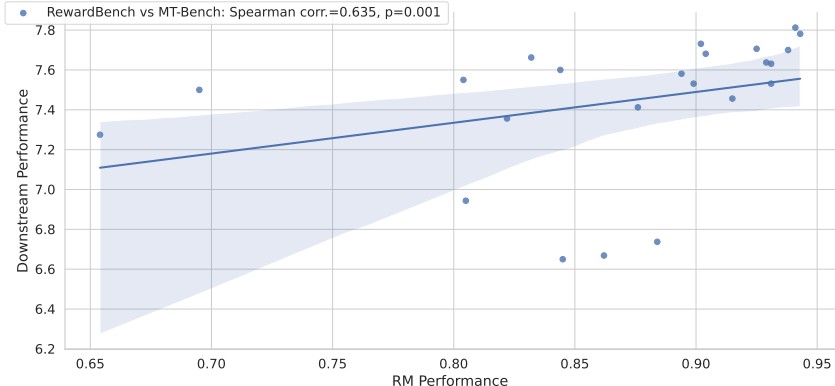

Figure 13: The correlation between RewardBench metric and MT-Bench score.

Table 7: The correlation between the accuracy and the policy regret with different KL penalties.

| KL Penalty | Relevance | | | |
|---|---|---|---|---|
| | Kendall $\tau$ corr. | Pearson corr. | Spearman corr. | MRR |
| 0.00 | 0.3980 | 0.6981 | 0.4840 | 0.5158 |
| 0.01 | 0.3743 | 0.5345 | 0.5011 | 0.4950 |
| 0.05 | 0.5487 | 0.7461 | 0.6692 | 0.3960 |
| 0.10 | 0.4925 | 0.6267 | 0.6265 | 0.3750 |

reward, reducing the likelihood that a policy optimized with a best proxy RM result in the best policy. Once the KL penalty exceeds a certain threshold, the KL reward begins to dominate in the later optimization stages, hindering further growth of the expected reward. This increases the influence of uncertainty in PPO optimization, leading to a decline in correlation. We also experiment with a larger KL penalty (KL Penalty = 0.5), finding that its effects could even cause training to collapse.

## 8.6 INFLUENCE OF SAMPLE SIZE

In Figure 6, we observe that increasing the number of responses improves correlation. However, it remains question of whether further expanding the sample size will continue to enhance correlation or not. Therefore, in this section, we examine how correlation changes with larger sample sizes. Since the original RewardBench contains only 2,733 different prompts, increasing the sample size to over 5,500 with just two responses per prompt can be infeasible. For sample sizes greater than 5,500, we only assess the correlation in cases with more than two responses per prompt, adjusting the number of responses accordingly for even larger sizes. From the result depicted in Figure 14, we find that further expanding the sample size does not significantly enhance correlation; rather, it gradually approaches an upper limit.

## 8.7 INFLUENCE OF DOWNSTREAM MODEL ON EXPANDING RESPONSE

In previous experiments, we use LLaMA-3-8b-Instruct as the downstream model and found that increasing the number of responses improved correlation with downstream performance. However, it remains questions of whether this effect holds with different downstream models. In this section, we use Qwen-2.5-7B-Instruct (Team, 2024) as the downstream model to test if increasing the number of responses still enhances correlation. As shown in the Figure 15, increasing the number of responses continues to be beneficial strategy with a different downstream policy.

## 8.8 REWARD ERROR METRICS

Given a prompt $x$ and $N = 5$ different responses $Y = [y_1, ..., y_N]$, their corresponding reward scores and ranks are represented as $r(Y)$ and $R(Y)$. Since each prompt has multiple annotated

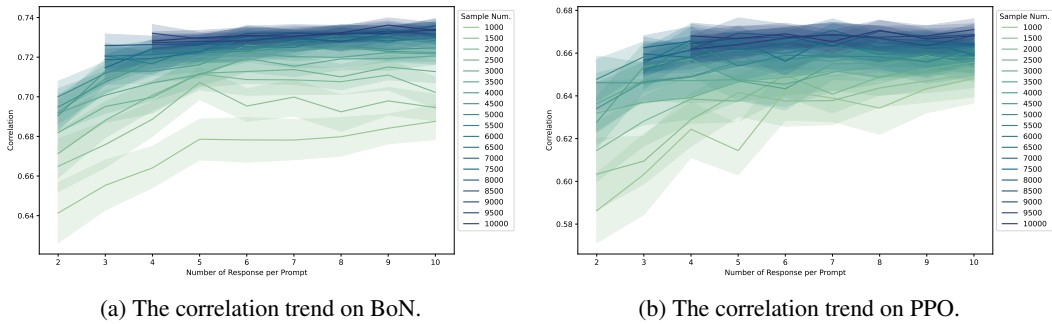

(a) The correlation trend on BoN.

(b) The correlation trend on PPO.

Figure 14: The correlation between accuracy and policy regret on test datasets with the same number of samples but varying numbers of responses per prompt assessed by the Spearman coefficient.

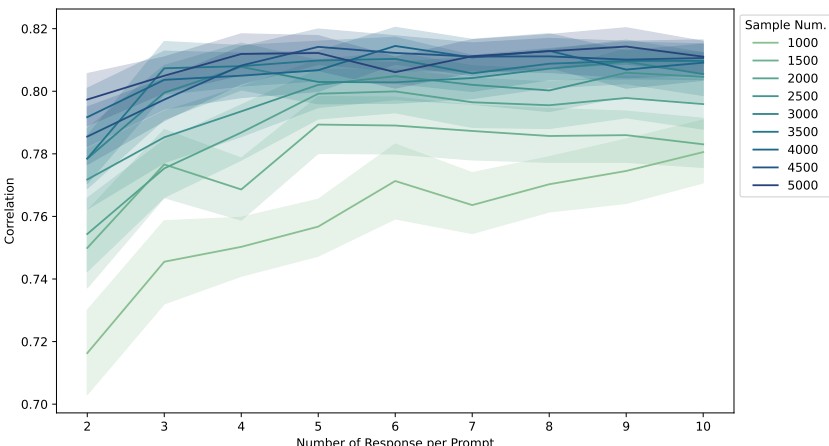

Figure 15: The Spearman coefficient between accuracy and policy regret with test datasets having the same number of samples but varying numbers of responses per prompt.

responses, many ranking evaluation metrics can be applied to our setting. Finally, we averaged the correlation metrics across all prompts to obtain the results presented in the Table 4.

**Pearson corr.** Pearson correlation coefficient can be computed as follows:

$$\rho_{r(Y),r^*(Y)} = \frac{cov\left(r(Y), r^*(Y)\right)}{\sigma_{r(Y)}\sigma_{r^*(Y)}} \tag{8}$$

where $cov$ is the covariance, $\sigma_{r(Y)}$ is the standard deviation of the proxy rewards given to the responses $r(Y)$ and $\sigma_{r^*(Y)}$ is the standard deviation of the golden rewards given to the responses $r^*(Y)$. The formula of $cov\left(r(Y), r^*(Y)\right)$ can be writed as:

$$cov\left(r(Y), r^*(Y)\right) = \mathbb{E}\left[(r(Y) - \mu_{r(Y)})(r^*(Y) - \mu_{r^*(Y)})\right] \tag{9}$$

Where $\mu_{r(Y)}$ and $\mu_r^*(Y)$ are the mean of the proxy rewards and golden rewards given to the responses, respectively. The final metric is the average of the coefficient of each prompt.

**Spearman corr.** Spearman's rank correlation coefficient follows the basically same formula as the Pearson correlation coefficient, only with the reward scores replaced by the ranks:

$$s_{R(Y),R^*(Y)} = \frac{cov\left(R(Y), R^*(Y)\right)}{\sigma_{R(Y)}\sigma_{R^*(Y)}} \tag{10}$$

**Kendall $\tau$ corr.** Kendall rank correlation coefficient is computed as follows:

$$\tau_{R(Y),R^*(Y)} = \frac{n_c - n_d}{N(N-1)/2} \tag{11}$$

where $n_c$ and $n_d$ stands for the number of concordant and disconcordant pair between $R(Y)$ and $R^*(Y)$, respectively. The concordant pair $(y_i, y_j)$ means that their rank satisfy $(R(Y_i) - R(Y_j))(R^*(Y_i) - R^*(Y_j)) > 0$. In practice, we employ $\tau_B$ which handles the ties for the rare case that some responses get the same reward scores. The final metric is the average of the coefficient of each prompt.

**Accuracy** The accuracy metrics are mostly the same as in the typical case that there are two responses per response. The main difference is that if there are $N$ responses per response, we can then form $C_N^2$ different pairs for comparison.

**Bo5** The best-of-5 metric can be seen as a special case of **NDR** for $N = 5$, which computes:

$$\frac{r^*\left(\arg\max_{y_i}\left[r(Y)\right]\right) - \mu_{r^*(Y)}}{\max\left[r^*(Y)\right] - \mu_{r^*(Y)}} \tag{12}$$

This metric represents the improvement in reward values obtained using the proxy reward score compared to those achievable with the original golden reward model. The final metric is the average of the coefficient of each prompt.

**ECE** Expected Calibration Error (ECE) is calculated by averaging the absolute differences between predicted probabilities and observed frequencies, typically across a set of bins partitioning the prediction space:

$$ECE = \sum_{m=1}^{M} \frac{|B_m|}{n} |acc(B_m) - conf(B_m)| \tag{13}$$

where $B$ represents the bins that split pairs by reward margins and $M$ stands for the number of bins. $acc$ is the accuracy of pairs in each bin. $conf$ computes the expected accuracy inferred from reward score margins by the Bradley-Terry Model (Bradley & Terry, 1952). Expected calibration error indicate the alignment of reward models' confidence. We follow the same strategy for form preference pairs as in the **Accuracy**.

**MRR** Mean reciprocal rank is a traditional Information Retrieval metric that can be transported to our setting smoothly. We first define reciprocal rank as the golden rank of the response that receives the highest reward score.[3] Then we take the average overall prompts:

$$MRR = \mathbb{E}_{x \in \mathcal{X}}\left[\frac{1}{R^*\left[\arg\max_{y_i}\left[R(Y)\right]\right]}\right] \tag{14}$$

**NDCG** Normalized discounted cumulative gain is another typical Information Retrieval metric that is transported here. This computes compute:

$$NDCG = \frac{\sum_{i=1}^{N}(R^*(Y_i) - 1)/\log_2(N - R(Y_i) + 1)}{\sum_{i=1}^{N}(R^*(Y_i) - 1)/\log_2(N - R^*(Y_i) + 1)} \tag{15}$$

The main difference from the typical usage in the field of Information Retrieval is that we replace relevance score $rel_i$ with the golden rank $R(Y_i)$. These metrics can be seen as a smooth version of **MRR**. The final metric is the average of the coefficient of each prompt.

---

[3]One may expect this metric to compute the proxy rank of the response that receives the highest golden reward score. But our implementation makes it more similar to the Bo5 metric.

$\xi$ **corr.** $\xi$ correlation coefficient (Chatterjee, 2020) is relatively new metric for evaluating the rank correlation. Compared to traditional rank coefficients like Spearman corr., this coefficient is more effective to compute. It first rearrange the data as $\left[r^*(Y_{(1)}), r(Y_{(1)})\right], ..., \left[r^*(Y_{(N)}), r(Y_{(N)})\right]$ such that $r^*(Y_{(1)} \leq r^*(Y_{(2)}) \leq ... \leq r^*(Y_{(N)}))$, and then compute:

$$\xi_N\left(R(Y), R^*(Y)\right) = 1 - \frac{3\sum_{i=1}^{N}|R_{(i+1)} - R_{(i)}|}{N^2 - 1} \tag{16}$$

## 8.9 THE RELATIONSHIP BETWEEN ACCURACY AND REGRET

Based on the assumption that only Regressional Goodhart takes effect and golden reward score $r^* \sim \mathcal{N}(0, \sigma_r^2)$, the noise $z \sim \mathcal{N}(0, \sigma^2)$, we can then derive the relationship between the accuracy and the regret. Based on the Regression Goodhart's, the proxy reward $r$ can represented as:

$$r = r^* + z \tag{17}$$

The process of constructing an RM test set can be viewed as performing two independent samples from the distribution of the golden reward score. Therefore, the scores obtained from the two samples can be represented as $r_1 \sim \mathcal{N}(0, \sigma_r^2)$ and $r_2 \sim \mathcal{N}(0, \sigma_r^2)$. Subsequently, the difference between the two samples' golden reward scores also follows a normal distribution: $r_-^* \sim \mathcal{N}(0, 2\sigma_r^2)$. Then the proxy reward model score difference can be written as:

$$r_- = r_-^* - z_1 + z_2 \tag{18}$$

where $z_1$ and $z_2$ is the noise introduced in the two times of sampling. The distribution of noise difference is also normal distribution $z_1 - z_2 \sim \mathcal{N}(0, 2\sigma^2)$. The accuracy can be translated into:

$$\begin{aligned}
d_r^{acc} &= P(r_- > 0, r_-^* > 0) + P(r_- < 0, r_-^* < 0) \\
&= P(r_- > 0, r_-^* > 0) \\
&= 1 - 2P(z_1 - z_2 < 0, r_-^* > 0) \\
&= 1 - 2\int_{x=0}^{+\infty} P(x = r_-^*)P(z_1 - z_2 < x)dx \\
&= 1 - \int_{x=0}^{+\infty} \frac{1}{\sigma_r\sqrt{\pi}}e^{-\frac{x^2}{4\sigma_r^2}}\Phi\left(-\frac{x}{\sqrt{2}\sigma}\right)dx
\end{aligned} \tag{19}$$

Following the notations in Section 2, we define the degree of overoptimization $d_\pi$ as follows:

$$d_\pi = \frac{J_{r^*}(\pi)}{J_r(\pi)} \tag{20}$$

This value implies that if the expected reward of a policy $\pi$ under the proxy RM is $\mathbb{E}(r) = c$, then the corresponding expected reward under the golden RM should be $mathbbE(r^*) = d_\pi c$. We then introduce a result from (Gao et al., 2022):

$$\mathbb{E}\left[X|X + Z = c\right] = \mathbb{E}[X] + (c - \mathbb{E}[X] - \mathbb{E}[Z])\frac{\text{Var}(X)}{\text{Var}(X) + \text{Var}(Z)} \tag{21}$$

where $X$ and $Z$ are independent, absolutely continuous random variables, both normally distributed. In our context, $X$ can be replaced by $r^*$ and $X + Z$ by $r$. Therefore, the degree of overoptimization

Table 8: Summary of Training Hyperparameters

| RM Training Hyperparameters | Value | PPO Training Hyperparameters | Value |
|---|---|---|---|
| Max Length | 2048 | Train Batch Size | 64 |
| Regularization Coefficient | 1e-2 | Rollout Batch Size | 8 |
| Batch Size | 256 | Generate Max Length | 1024 |
| Warmup Ratio | 0.1 | Actor Learning Rate | 1e-6 |
| Learning Rate Scheduler | cosine | Critic Learning Rate | 1e-5 |
| Learning Rate | 5e-6 | KL Penalty | 0 |

$d_\pi$ can be expressed as:

$$
\begin{aligned}
d_\pi &= \frac{\mathbb{E}_{y\sim\pi(\cdot|x)}[r^*(x,y)]}{\mathbb{E}_{y\sim\pi(\cdot|x)}[r(x,y)]} \\
&= \frac{\int_{-\infty}^{+\infty}\int_{-\infty}^{+\infty} r^* p(r^*,r)dr^*dr}{\int_{-\infty}^{+\infty} rp(r)dr} \\
&= \frac{\int_{-\infty}^{+\infty}\left[\int_{-\infty}^{+\infty} r^* p(r^*|r)dr^*\right]p(r)dr}{\int_{-\infty}^{+\infty} rp(r)dr} \\
&= \frac{\int_{-\infty}^{+\infty} \frac{r\sigma_r^2}{\sigma_r^2+\sigma^2}p(r)dr}{\int_{-\infty}^{+\infty} rp(r)dr} \\
&= \frac{\sigma_r^2}{\sigma_r^2+\sigma^2}
\end{aligned}
\tag{22}
$$

Under the BoN setting, we first normalize the reward scores and directly estimate $d_\pi$ according to the definition in equation 20.

## 8.10 Details about Experiments

For all experiments in Section 4, we obtain the results through multiple samplings of test datasets. Specifically, in Figure 4, we perform $64$ samplings for each chosen-reject rank bin to construct different test datasets and calculate their correlation with downstream performance. The final results are averages of these outcomes. For Tables 2, 3, and 4, the set of prompts is fixed, but we sample multiple different responses for each prompt, conducting $64$ samplings to construct various test datasets. These outcomes are used to calculate the mean and variance for reporting correlations. In Figures 5, 6, and 7, neither the test prompts nor responses are fixed. For each data point, we perform $64$ samplings to obtain different test datasets and calculate the results, which are reported with the 95% confidence interval.'1 Next, we provide training details. The hyperparameters for RM and PPO training are listed in Table 8. For PPO optimization, we utilize the OpenRLHF framework (Hu et al., 2024). In BoN optimization, $N$ is set to 1280, requiring $2733 \times 1280 \times 10 = 34,982,400$ response samplings and RM scoring for preparing BoN results. In the KL divergence calculation for BoN, we use a logarithm base of 2.

