# OpenReview forum: "Rethinking Reward Model Evaluation: Are We Barking up the Wrong Tree?"
_ICLR.cc/2025/Conference — ICLR 2025 Spotlight_

### Official Review · Reviewer_FwA4 · 2024-10-30

**Soundness:** 2
**Presentation:** 2
**Contribution:** 2
**Rating:** 5
**Confidence:** 3

**Summary:**

This paper empirically investigates whether "reward model (RM) accuracy" (i.e., agreement with a preference dataset), which is commonly used for evaluating reward models, is a good metric for evaluating reward models. It begins with the premise that the true measure of a reward model is the quality of the policies it produces, and proceeds to investigate the relationship between RM accuracy and the regret of the downstream policy. It does this by synthesizing several "noisy" reward models, which can each serve as both a "gold" reward model providing ground truth labels for both preferences and policy performance, or a proxy reward model (when another RM is the gold reward model). The authors then proceed to conduct several empirical investigations into the RM accuracy vs Policy regret. They conclude that (1) RM accuracy is not perfectly correlated with downstream policy regret, (2) the predictive power of RM accuracy declines as the distribution on which policy regret is measured shifts away from the original distribution, and (3) ultimately conclude that RM accuracy is not adequate as a measure of RM quality.

**Strengths:**

The research question is a good one. That is, it is known from the RL literature that the best reward models for learning are not necessarily the ground truth reward models (e.g. Singh et al. 2009, Where Do Rewards Come From?), so it seems natural that some proxies of ground truth rewards (well, in case of LLMs, trajectory returns) will be better than other proxies of ground truth rewards --- can we ascertain whether this is the case, and if, so can we find ways to craft better proxies and/or better measures of proxy quality. I think this is an important question of great interest to a variety of researchers. I also think it has not really been studied in the case of language reward models, so it is good to have a paper that prompts the discussion.

To answer this question, this paper contains several experiments that show the relationship between RM accuracy and downstream policy performance. The questions that drive the experiments are generally interesting. There are a lot of experimental outcomes here, which may be interesting to different researchers. Although some thinking is necessary on part of the reader to understand what the authors are doing, the paper is fairly written through Section 4.

**Weaknesses:**

Issue: The fact that your hyperparameters for training the policy are held constant does not imply that the policy extraction / learning process is not noisy. So any characterizations of the correlation as being "weak" (L63, L530) or "room for enhancing" (L213) are IMO unsubstantiated, as the upper bound for correlation may be much lower than 1 due to variance in the policy optimization process. This noise also explains why correlation with PPO is weaker than correlation with BoN (which you correctly point out at L190). *So the major issue I have is that, even after reading this paper, I expect RM accuracy to be a better measure of expected downstream policy quality than single seed downstream policy quality.* (and this is the main cause for my low review score; the experiments do not convince me that RM accuracy is an insufficient target for designing RMs)

Although the experiment at L246-256 / Figure 4 is interesting, the descriptions are insufficient to properly understand Figure 4 (which axes is which, how does (a) map on to what you wrote in the text?). Further, what is the practical import of this experiment --- would we need humans to rank multiple samples for this to be relevant? And if so, wouldn't it then be better to just use all the samples as you investigate later on?

Finding 4... it would be good to provide an intuition for the reader as it was not immediately obvious to me why this is happening. Namely that the number of comparisons (and therefore samples with which to reduce variance) grows factorially in the number of responses. Once stated this way, I would note that past work has used multiple responses/ human labels per prompt, see e.g., Hwang et al. 2023, Sequential Preference Ranking for Efficient Reinforcement Learning from Human Feedback and Wu et al. 2023, Fine-Grained Human Feedback Gives Better Rewards for Language Model Training. (This is not to say Finding 4 is not interesting, but I would still rather see an analysis of how the annotation budget affects the RM accuracy, as opposed to the downstream Policy Regret). This all being said, perhaps if this finding were fleshed out more, to show how you can use it to improve RewardBench, it could be a lot more impactful.

Section 5: I'm afraid I do not really understand this section. I'm not sure the Goodharting / overoptimization / exogenous variable language is doing much good here (what would be an example of an exogenous variable impacting the relationship, and why is implied by the experiments here?). It seems you are assuming that rewards follow a Thurstone Choice model (whereas most literature assumes Bradley-Terry-Luce; which is fine, they are very close), and then in Figure 8(a) comparing the percentage of reward variance owed to the ground truth reward (vs noise term) to the RM accuracy. What is Figure 8(b) showing? How do we know what $d_\pi$ is in the "actual" setting? I don't understand what the takeaway from Figure 9 should be, if we can't see the reward accuracy, and don't know the sample size under which it computed, etc. There is detail missing here.

Important (but minor review-wise):
L139: alpha is never specified (except in Fig 9)
L159: n is never specified!
And generally, I think there are details left out / I would not be able to reproduce everything given current manuscript; e.g. what hparams were used for PPO, etc. How are confidence bands in Table 2 computed, etc.

Minor:
- L141: N x (N-1) pairs
- L48: "human golden reward function" --> "empirical human rewards"  (the human reward 'function' is noisy, and we cannot quantify the error between learned RM and the human reward fn)
- L50: "the goal of RM eval is to estimate policy regret" (no, it's to estimate the quality of the reward model --- policy optimization introduce a whole range of additional noise / issues -- see commentary above)
- L429: I would not say the entire translation is due to overoptimization... in fact, there may be NO overoptimization if we do early stopping, right?

**Questions:**

1. Do you have any potential solutions for the issues you have identified (how do we go beyond RM accuracy)?

2. Can you clarify if I am missing something re: my major concern above, or make me doubt the view that "RM accuracy is a better measure of expected downstream regret than single seed downstream regret". An experiment that shows insufficient correlation given multiple seeds might do the trick (even if you can't do PPO, you should be able to run this experiment quickly in the BoN setting).

---

> ### Author Response · Authors · 2024-11-23
>
> Thank you for the detailed feedback. We address your concerns below.
>
> *Responses to main weakness:*
>
> 1. **Regarding your main concern**, we want to clarify our statements. In this work, we are not aiming to conclude that accuracy is an unreliable metric. Rather, we found that it shows a certain level of correlation with downstream performance. We have explored various factors that affect this correlation and discussed potential strategies to enhance it. Our results suggest that preparing multiple responses for each prompt, rather than the common practice of including only two responses, would be beneficial. However, as demonstrated in Figures 5(a) and 5(b), even within similar terms of accuracy, different overoptimization behaviors can be exhibited. Moreover, several concurrent studies [1,2] noted a weak correlation between RM accuracy and certain downstream tasks, which we believe partially supports our conclusions. Moreover, we examine the correlation between accuracy (RewardBench Score) and actual downstream task performance (MT-bench score) across diverse RMs, as shown in Figure 13. The results indicate that the RewardBench rankings are not fully maintained in downstream tasks, supporting our statements.
>     - [1] RMB: Comprehensively Benchmarking Reward Models in LLM Alignment
>     - [2] Evaluating Robustness of Reward Models for Mathematical Reasoning
>
> 2. **Regarding the takeaways from the response rank experiment**, we would like to clarify that the objective of this experiment is not to directly find a strategy for constructing test sets, but rather to explore the potential influencing factors. In Finding 2, we first examined the impact of the model used for sampling responses. We discovered that sampling from different models results in different correlations. We hypothesize that the distribution of response reward scores (indirectly represented by response rank) may cause this phenomenon, and we conducted experiments to verify this. Although response rank itself is not a direct method for constructing test sets, it informs our choice of models for sampling responses. For stronger models (e.g., GPT-4o, GPT-4), we would expect higher ranks, while for weaker models, lower ranks. Based on the results of the response rank experiment, we can better choose the sampling models to improve correlation when constructing benchmarks.
> 3. **Regarding the experiments regarding the number of responses**, we greatly appreciate your suggestions. The main finding of this section is that increasing the number of responses per prompt in the test dataset is effective in most settings. This suggests that, when constructing the RM benchmark, it is beneficial to prepare multiple responses for each prompt, rather than the current practice of having only a chosen and a rejected response. This approach aims to achieve better correlation with downstream tasks. We hope this result can serve as a reference for future related work.
> 4. **Regarding the content confusion about Section 5**, we would like to provide further clarification. In this section, we assume normally distributed reward scores and noise to theoretically derive the relationship between accuracy and the degree of optimization $d_\pi$. We have provided a more detailed definition and derivation of  $d_\pi$ in Appendix 8.9. In the BoN scenario, $d_\pi$ can be directly estimated based on its definition. Comparing Figures 8(a) and 8(b), we found there are many outliers in the BoN scenario. We then analyze the sources of these outliers in Figure 9, where we have included additional accuracy data of these golden-proxy RM pairs for reference. Despite similar accuracies, some pairs display significant differences in overoptimization behavior. This suggests that accuracy alone cannot fully predict potential downstream overoptimization. We have revised the relevant content to reduce understanding difficulties.
> 5. **Regarding the suggestions on expressions and writing**. We try our best to address the issues you mentioned, along with other potential errors. Additionally, we have added more experimental details in Appendix 8.10. We would also like to further clarify the RM overoptimization question. Following the definition by OAI [1], we view RM overoptimization as a result of the Goodhart effect. Specifically, the regressional Goodhart effect occurs when the proxy reward function is essentially the golden reward function mixed with random noise. In such cases, optimizing the proxy RM to a specific value (e.g., r=10) leads to the expected reward $E(r^*)$ of the golden RM being lower than r=10. Early stopping cannot resolve this issue since it is not caused by optimization noise. However, we acknowledge that the transformation from RM error to policy regret is not entirely due to overoptimization. We have revised overly absolute statements accordingly.
>     - [1] OpenAI. Scaling Laws for Reward Model Overoptimization.

---

> > ### Author Response · Authors · 2024-11-23
> >
> > *Response to the questions:*
> >
> > - **Q1:** In addition to enhancing data construction methods to improve accuracy correlation, future work may focus on analyzing the preference patterns learned by RMs using RM interpretability techniques. In our experiments, we found that the characteristics of RMs can significantly influence both the BoN and PPO optimization processes. Some RMs are easier to optimize, achieving higher reward gains with the same KL, while others generalize better, enabling reward gains on the proxy RM to translate more effectively to the golden RM. Through deeper analysis of RMs, one can better predict RL training dynamics and potential overoptimization phenomena, which we believe cannot be predicted solely by evaluating accuracy. For instance, Figure 9 shows that despite some proxy-golden RM pairs having similar accuracies, BoN optimization behaves quite differently in terms of overoptimization.
> > - **Q2:** Thank you for your question. We addressed this question in response to the main concern.

---

> ### Author Response · Authors · 2024-11-24
>
> Dear Reviewer,
>
> Considering the rebuttal deadline is approaching, we sincerely hope to receive your response. If you have any further questions or concerns regarding our explanation, please do not hesitate to contact us. Your response is of great significance to us for improving this work, and we look forward to hearing from you.

---

### Official Review · Reviewer_UcNB · 2024-10-31

**Soundness:** 3
**Presentation:** 3
**Contribution:** 4
**Rating:** 8
**Confidence:** 4

**Summary:**

The paper questions the current approach to evaluating reward model performance, which is based evaluating downstream policy performance and looking at reward model accuracy on the train and eval preference datasets. The relationship between reward model performance and policy performance is explored in the context of policy regret, a new performance metric that examines the difference in policy performance when trained with the true versus inferred reward. Since the true reward is not typically available, a synthetic true reward model is used to conduct the analysis. The reward model and policy performance relationship is measured as the correlation between the reward model's accuracy and the policy regret. Evaluation relies on RewardBench. The authors find a weak positive correlation between the two, and conclude that accuracy alone is a poor proxy for downstream performance.

**Strengths:**

* Understanding the relationship between our measures of reward quality and policy quality is of vital importance.
* The goal and motivation for the work is well laid out at the start of the paper. The takeaways the reader should expect to have are presented from the start.
* The use of a synthetic ground truth reward function is well motivated and contextualized.
* Different methods for using a ground truth reward function (best-of-n and RL) are compared.

**Weaknesses:**

**High level**

The main weaknesses for this paper are not overly large and mostly involve clarifications to the text. While not big changes, they are important to address. Some of the conclusions in the main body need to be walked back and made more nuance to fully reflect the presented results. The biggest missing result is information about the relationship between the ground truth reward model and the proxy reward model.

**Details**

* The experiments are set up such by taking a labelled preference dataset and then randomly flipping labels to create multiple preference datasets. A different reward model is then trained on each version of the dataset. The ground truth reward model is then arbitrarily chosen to be one of those and all others are treated at the proxy. It would be good to quantified the relationship between the ground truth and proxy reward models as part of the analysis. This could be as straightforward as looking at the percentage of agreeing labels in the training data.
* It would improve understanding of the experimental section to describe the RewardBench data and how it is used earlier. The very start of Section 3, would be a good place.
* It would  strengthen the results and analysis to include policy regret information where the ground truth reward is used as the proxy reward. This would help to understand the impact of randomness in policy learning. In practice, is the policy regret 0?
* Some of the statements made early in the paper (e.g. introduce) are difficult to interpret in the absence of having read the whole paper. For example, "...we find that  the rank of the responses can be more influential..." and "...increasing the number of responses per prompt". It would be helpful to add some description about what is meant by rank and what it means to increase the number of responses in terms of the evaluation.
* It is challenging to interpret the results in Figure 4. Adding axis names would be beneficial.
* The conclusion that "While the correlation on PPO continuously weakens as we paragraph a larger number of prompts", the nuance that this depends on the correlation metric should be called out, especially as no single correlation measure has been identified as "best".
* Please add more descriptions of the different methods used to evaluate Finding 4. Specifically things like "Bo5" where a citation would also be helpful.
* It is not clear how all of the metrics in Table 4 were computed. For example, what is the Pearson corr. measured between?
* The conclusion about the impact of number of responses per prompt does not fully reflect the results in Figure 7 (a). For the smaller annotation budgets, the benefit of extra responses drops off quickly. The trends for PPO should be summarized and described and extra responses are not beneficial.
* The paragraph immediately after "Finding 2" (lines 238 - 244) were not clear to me, so it is difficult for me to assess or validate them. Parts such as "solely from different models" was not clear. It is not clear exactly how the data is different from what was used previously.
* Some small spelling issues, typos, and confusing phrases throughout the paper. These are not impacting my score, would be good for the authors to clean up.
     * line 071 "investigate" -> "investigating"
     * line 089 "This offers valuable for both reward model training...."
     * line 129 "...we focus on a few interested factors and keep the rest fixed" - "interested" -> "interesting"
     * line 134 "...perform investigation..."
     * line 319 "...expect calibration error..." -> "...expected calibration error..."

**Questions:**

* How do the results here relate to the findings in "Preference Learning Algorithms Do Not Learn Preference Rankings" (NeurIPS 2024)?
* Where does the data used to evaluate the policy and measure policy regret come from?
* How do the results about prompt versus response distributions relate to the paper "On the Limited Generalization Capability of the Implicit Reward Model Induced by Direct Preference Optimization" (Lin et al., 2024)?
* It seems the paper does not account for noise. Why is that not important here?
* In equation (2), what is $\pi_{0}$? Is this the SFT/IFT'ed model?
* What is your motivation for setting the KL penalty to 0 (line 160)?
* For the analysis of Figure 3 supporting the conclusion that  "policy regret can very considerably even within similar accuracy ranges", is it possible that accuracy and NDR exist on different scales making it look like there is more variation along one dimension that the other? Can you report the results normalized so that they fall on the same scale?
* For the results reported in Figure 3, how OOD versus ID is the data for each the reward model and the policy?
* How were the bins that were used to assign response rankings determined? Was the ground truth reward model for that experiment used?
* It is surprising that seeing higher reward samples was not more beneficial. Can you elaborate more on why this is the case?
* For Figure 4, what does it mean when the bin is 0 for both the x and y axis? Does it mean the two responses are of equal rank? In this case, how does labelling work?
* For the results looking at the impact of response rank, how was the label flipping or disagreement with the ground truth reward function's training data accounted for? For the flipped labels spread uniformly over the rank bins?
* Why is Table 3 not symmetric along the diagonal?
* What is the hypothesis for why extra responses help in the case of BoN and not PPO?
* In "We observed that there should be an approximate linear relationship between them" (line 454), "observed" + "should be" in unclear. You assume? You know? Also how? Why?

---

> ### Author Response · Authors · 2024-11-23
>
> Thank you for the positive feedback and the constructive comments. We address your concerns below.
>
> *Responses to main weakness:*
>
> - Thank you for your valuable feedback. We have clarified and enriched the text to convey our findings more accurately and effectively. Regarding the missing results, we have added Appendix 8.3 to illustrate the relationship between golden and proxy RMs. This section presents the accuracy between the golden and proxy RMs, as well as the NDR metric relationships between proxy-golden pairs under BoN and PPO optimization. We observed that accuracy generally increases as data noise decreases, and extreme NDR values are more likely to occur under PPO optimization.
>
> *Responses to detailed weakness:*
>
> - **W1:** Regarding the agreeing labels, we have added an explanation in the main text. When constructing the data, we ensured that if a training set has $a_i$ of its labels flipped and another training set has $a_j$ flipped, then exactly $|a_i-a_j|$ of the labels between these two training sets will be inconsistent.
> - **W3:** According to the definition of NDR, we measure the difference in expected reward when optimizing towards a proxy RM compared to a golden RM under the same hyperparameter settings. Therefore, when using the Golden RM as the Proxy RM, this value is equal to 1 by definition.
> - **W7,8:** We have detailed the calculation methods for the various metrics in Finding 4 in Appendix 8.8. Specifically, each prompt's five responses are scored by both the golden and proxy RMs, allowing us to rank these responses accordingly. For instance, using Pearson correlation, if the golden RM scores the five responses as [0.1, 0.2, 0.02, -0.11, 0.3] and the proxy RM scores them as [0.12, 0.22, 0.12, 0.15, -0.3], the correlation coefficient is -0.6091. We compute these correlations for all prompts, and the final result is obtained by averaging them.
> - **W9:** Thank you for your suggestion; we have improved the associated content. We discuss this topic further in response to Question 14.
> - **W10:** Apologies for the confusion. In this paragraph, we explore whether it is necessary to sample responses from the model used for downstream optimization when constructing the test dataset. We specifically build multiple test datasets, each involving responses sampled from only one model. We found that using the downstream policy for sampling responses is unnecessary. We have revised the content to clarify this point.
> - **Regarding the remaining suggestions to enhance the clarity of the article**, we revised the corresponding sections you pointed out and made further adjustments throughout the paper to enhance its overall readability.

---

> > ### Author Response · Authors · 2024-11-23
> >
> > *Response to the questions (1-7):*
> >
> > - **Q1:** The article you mentioned primarily explores the issue that RLHF or DPO optimization algorithms may not accurately learn preference rankings. This work points out the problem present in current RL optimization algorithms. The findings of this paper suggest that even if RL algorithms optimize towards the Golden RM, they may still struggle to fully fit the distribution represented by the Golden RM. This is partly related to our statement that the ideal regret defined in Eq(2) cannot be practically computed, as these results imply that obtaining a globally optimal policy under a given RM is challenging. Therefore, in our work, we use the NDR metric to minimize the influence of the optimization algorithm.
> > - **Q2:** In this work, we use prompts from the RewardBench dataset to test policy regret. We've added relevant explanations at the beginning of Section 3 to avoid further confusion.
> > - **Q3:** The paper you mentioned investigates the differences in generalization between traditional RMs and the implicit RM represented by DPO, concluding that DPO's implicit RM tends to generalize less effectively. This paper focuses on the generalization issues of RL algorithms, highlighting that even if current RL algorithms perform well on the RL training set, they may experience varying degrees of performance decline on the test set. In our work, we investigate the relationship between RM accuracy and downstream policy regret on test sets from different distributions. We believe that this primarily relates to RM generalization; if an RM generalizes well, we would expect the correlation between accuracy and downstream policy regret to be consistent across different test distributions.
> > - **Q4:** Our approach of adding noise to the training data to an extent accounts for the common types of noise encountered in real-world scenarios. Conversely, noise arising during RM training and RL optimization is more challenging to quantify and control. Therefore, we mitigate their impact as much as possible by increasing the sample size of proxy-golden pairs.
> > - **Q5:** Apologies for the confusion; $\pi_0$ denotes the initial policy, which is the IFT model (Llama3-8b-Instruct) in our experiment. We have added further descriptions to clarify this and avoid further misunderstandings.
> > - **Q6:** For setting the KL penalty to 0, we followed the setup from the classic OAI paper (https://arxiv.org/abs/2210.10760). Additionally, we included experiments on the impact of KL in Table 7. From our experience, the KL coefficient affects the optimization behavior of the PPO algorithm. When the KL coefficient is large, the KL reward may dominate the optimization process, suppressing the increase in the proxy RM. In our experiments, we found that with a high KL coefficient (up to KL=0.5), the PPO optimization process can collapse in the later stages. Since our experiments primarily focus on the potential overoptimization observed with a proxy RM relative to the golden RM, additional KL constraints could hinder the improvements in the proxy RM, so we set it to 0.
> > - **Q7:** Regarding the scales of the Accuracy and NDR metrics, accuracy ranges from 0 to 1, while NDR theoretically spans from  $-\infty$ to $+\infty$. This scale difference makes normalizing NDR challenging. However, by definition, if the accuracy between the proxy and golden RM exceeds 0.5, we would expect an NDR value between 0 and 1, with the NDR approaching 1 as accuracy increases. This is because higher accuracy indicates greater similarity between the proxy and golden RMs, suggesting that a policy optimized using the proxy RM should perform comparably to one optimized with the golden RM. Nonetheless, as shown in Figure 4, despite observing positive correlations in both BoN and PPO, NDR values can vary considerably for golden-proxy pairs with similar accuracy. For instance, under BoN optimization, when accuracy is around 0.85, NDR can fluctuate between approximately 0.2 to 0.6. And this fluctuation is even more pronounced under PPO optimization.

---

> > > ### Author Response · Authors · 2024-11-23
> > >
> > > *Response to the questions (8-15):*
> > >
> > > - **Q8:** In Figure 3, all RMs are trained on the same RM training set and subsequently tested on the RM test set. The policy is evaluated on the RL test set, which is the same as the RM test set.
> > > - **Q9:** Each RM takes a turn as the golden RM to determine the rank of responses and construct different test datasets accordingly. We then calculate the accuracy of these test datasets and assess the correlation with the corresponding policy regret.
> > > - **Q10:** For the results in Figure 4, we observed that the most effective samples vary between algorithms (BoN and PPO). For PPO, it is beneficial to use high-reward samples as positive samples and mid-reward samples as negative samples. Conversely, for BoN, it is advantageous to use mid-reward samples as positive samples and low-reward samples as negative samples. This difference may relate to the characteristics of the two algorithms. With PPO, the policy is optimized more aggressively, so the responses tend to rank higher under the golden RM. This makes the consistency between the proxy RM and the golden RM on high-reward samples more critical. In contrast, for BoN, where the level of policy optimization might not be as intense, the consistency on mid-reward samples can be more instructive. We have added additional explanations to the relevant paragraph in the revised version.
> > > - **Q11:** We have re-plotted Figure 4 and provided additional descriptions. In cases where both the x and y-axis bins are set to 0, the chosen and rejected samples are both selected from those ranked between 0 and 5. We ensure that the rank of the chosen responses is higher than that of the rejected responses.
> > > - **Q12:** The random flipping of labels is performed on the training set, which allows us to train different RMs. In the response rank experiment, there is no label flipping in the test set. The differences between test sets lie in the rank of the responses selected.
> > > - **Q13:** In Table 3, we observe that the BoN algorithm's policy and regret show a stronger correlation on prompts from the same distribution, whereas this correlation appears more random for the PPO algorithm. This may be due to the inherent characteristics of the PPO algorithm. It can be easier to achieve reward gains for certain types of prompts. Consequently, the optimized policy might exhibit higher KL divergence in these categories, resulting in greater reward improvements. This imbalance in optimization across different response categories could be the cause of the phenomena.
> > > - **Q14:** With the same number of samples, both BoN and PPO show stronger correlations as the number of responses increases. This is reasonable because more responses per prompt lead to more pairs being formed, thereby enhancing correlation. However, when controlling for the annotation budget, we consider the annotation noise, which can render the test set unreliable. This noise may not significantly affect BoN due to its inherent stability. In contrast, PPO, being a less stable optimization process, is more susceptible to the effects of test set noise.
> > > - **Q15:** Apologies for the confusion. In Figure 8, we want to compare the actual trend under the BoN setting to the expected trend derived from Eq(7). We have revised the corresponding expressions.

---

> ### Comment · Reviewer_UcNB · 2024-11-23
> **Response to Author Rebuttal**
>
> Thank you for your response and the corresponding changes to the paper. I am adjusting my score.

---

### Official Review · Reviewer_22rE · 2024-11-01

**Soundness:** 3
**Presentation:** 3
**Contribution:** 3
**Rating:** 8
**Confidence:** 4

**Summary:**

This paper revisits the soundness of current reward model evaluation procedures, which assess the accuracy of RMs over datasets of held-out preference data, showing they present the flaw not to necessarily translate into downstream improvement for models going through RLHF in a synthetic setting. They propose to improve RM evaluation by increasing the number of responses per prompt and/or (if available) using the rank of model responses to select preferred and rejected responses when building preference datasets.

**Strengths:**

The problem tackled is well introduced and presented.

Contributions are clear.

Having a synthetic and cheap setting that correlates with more realistic experiments is quite valuable for the research community (but see doubts expressed below).

Some very interesting findings in this setting:
* weak correlation between RM accuracy and downstream policy performance/regret
* correlation increase by increasing the number of answers per prompt
* correlation increase by picking answers based on their rank
* correlation increase by increasing the number of test samples

EDIT: I raised my score accordingly after convincing updates from the authors.

**Weaknesses:**

The key limitation from my perspective is that we have currently no solid evidence that the study will translate to real settings: diverse, heterogenous RMs potentially trained on different datasets. While this would be infeasible to run as many experiments and ablations as in the synthetic setting, showing that there is imperfect correlation between RewardBench scores and relevant, downstream RLHF settings (more generally, that some findings from the synthetic study do replicate) would make the paper much stronger. At least showing that RewardBench rankings are not preserved after downstream BoN / RLHF would be a great addition.

In that regard, the fact that RMs used in the experiments are essentially the same architecture / same init / same hyperparameters trained on very similar data (i.e. up to random flips) is concerning: this is not capturing the diversity of RMs being trained and evaluated on popular benchmarks such as RewardBench. Authors should do a better job at showing that this is enough to capture phenomena that replicate in real settings.

Also, what about using a larger RM as proxy-golden? Seems like the default practice and would make for a more principled study, as this would likely reduce RM similarity with the proxy-golden RM.

Another limitation is that it is currently hard to evaluate whether the correlation increase authors get from applying the proposed improvements are significant or not. What is the correlation between proxy-golden RM accuracies on train and test data? This value would constitute an upper bound of what level of correlation is realistically achievable and provide a reference value that would alleviate the above concern.

Several ablations are missing AFAICT:
* Why is correlation higher for BoN vs PPO? Authors state that “This is expected, as BoN is a more localized and stable optimization algorithm, making it more predictable by reward model error.” I think this might be due to using unconstrained PPO (i.e. without KL regularization), which begs for an ablation study.
* Did authors conduct any form of investigation on low accuracy but high NDR PPO policies from Fig. 3 b)?
* It is currently unclear whether adding more samples (i.e. more than 5000 samples) in Fig 6 would improve correlation or instead saturate? This experiment would be a compelling addition to the study.
* Regarding using additional responses per prompt: the question of whether the policy matches or not the downstream policy matters or not is not studied in the current state
* What about using a different set of prompts to quantify whether the difference between test RM prompts and test RL prompts is meaningful?

Notations and equations:
* equation 2 uses $\pi’$ on LHS but not present on RHS -> please fix
* what is \pi_0 in equation 2? I suppose it is the initial policy, but this should be clarified in the main text
* is the regularization term from Equation 3 known/standard? if so, clarifying and including a citation to prior works is warranted

Clarity can be improved a lot:
* “on the widely adopted benchmark dataset (Lambert et al, 2024)” -> authors should name it (RewardBench)
* “Regarding response distribution, we find that the rank of the responses can be more influential rather than the model from which it is sampled.” -> unclear sentence
* “The translation from the RM error to the policy regret can be seen as the result of the reward model overoptimization” -> “imperfect translation”? or “weak correlations”? unclear sentence for now
* “ the difficulty of constantly controlling optimization pressure” -> what do the authors mean by pressure here?
* is the Normalized Drop Ratio (NDR) a contribution of the authors’ work? if so this should be clarified in the text
* the use of NDR (as opposed to difference in average rewards) should be motivated better in the text, even if easy to interpret
* Fig 3 is a bit hard to read (dots / text could be made bigger)
* /!\ Using more responses per prompt to improve RM evaluation -> This is a key aspect of the paper that is quite unclear in the current state! As of now the details are in the appendix, but the main text should be much clearer about this aspect, notably that the fact that we have a golden-proxy RM allows us to estimate a correlation between the proxy rewards and the golden-proxy rewards.
* “Can we achieve a higher correlation by sampling responses specifically from the model used for downstream optimization? To examine this question, we construct test datasets with responses solely from different models.” -> I find these sentences puzzling as they seem to contradict each other, please clarify

Writing is sub-par in the current state, see examples:
* “The inherent difficulty of constructing an ideal RM require”
* “The latter, while straightforward, remains the question of whether such evaluation accurately predicts the performance of the downstream optimized policy”
* “A cartoon of RM error and policy regret”
* “We begin by investigate the influence of prompt and response distributions”
* “The correlation between policy regret and accuracy on datasets constructed from responses of different ranks assessed the by Spearman coefficient”

with more in the text.

**Questions:**

See questions above as well.

Fig 3 shows high correlation between accuracy and BoN perf at high accuracy. Do authors have an intuition on why that might be the case? Also, no RMs have an accuracy between 0.75 and 0.8, which might indicate a lack of diversity in the RMs trained, echoing an earlier remark on experimental design.

“As shown in Table 2, this approach does not consistently improve upon the original RewardBench dataset. This result suggests that sampling responses from the model used for optimization may not be necessary to achieve a strong correlation between accuracy and policy regret.” -> this is quite surprising. Do authors have an intuition here?

---

> ### Author Response · Authors · 2024-11-23
>
> Thank you for the constructive feedback. We address your concerns below.
>
> *Responses to main concerns:*
>
> 1. **Regarding the concern about the lack of diversity in the RMs** used in our study, we recognize the importance of exploring this aspect. To this end, we have conducted additional experiments, detailed in Appendix 8.4.
> First, we assess the correlation between accuracy and policy regret in different RM synthetic settings, including training RMs with different models and on different datasets.  Second, we evaluate the correlation between the RewardBench scores of diverse RMs and downstream task performance using MT-Bench evaluated with GPT-4o, with the optimization of LLaMA-3-Instruct via Best-of-32.
>
>     These results highlight an imperfect correlation between RM error measurement and downstream performance. Furthermore, some concurrent works [1, 2] investigated the relationship between RewardBench and various downstream tasks. They observed a similar weak correlation between RewardBench results and downstream performance.
>    - [1] RMB: Comprehensively Benchmarking Reward Models in LLM Alignment
>    - [2] Evaluating Robustness of Reward Models for Mathematical Reasoning
>
> 2. **Regarding the use of larger RMs as proxy-golden models,** we would like to clarify that our approach involves preparing *N* RMs and pairing them to form $N(N-1)$ Proxy-Golden pairs for correlation assessment, as illustrated in Figure 2(b). In this setup, designating a single larger model as the Golden can be challenging due to the nature of pairwise comparisons. Meanwhile, we acknowledge that the size of RMs is a critical factor to investigate. To address this, we conducted additional experiments using 14 RMs of varying sizes (trained on 0.5 to 72 billion parameters, including pre-train and instruct models from the Qwen 2.5 series) in Figure 12(a). In Figure 12(a), we also find many outliers, suggesting that RM rankings are not fully preserved in their downstream performance.
> 3. **Regarding the significance of our results**, we included variance indicators in most of the tables and figures in Section 4. All results are derived by averaging over multiple rounds of random sampling for building the test dataset,  to ensure the robustness and consistency of the findings. Details of the experimental setups are provided in Appendix 8.10.
> Additionally, regarding the correlation between the accuracy on the training and test sets for proxy-golden models, we sampled a subset from the training set of the same size as the test set. We then calculated the accuracy of the proxy-golden model on both the training subset and the test set, and measured their correlation in the Table. However, while this value may relate to the correlation between RM performance and downstream results (if the RM test dataset prompts differ from the RL test dataset prompts), we think it does not necessarily establish an upper bound. This is because the correlation you mentioned reflects the generalization of RMs from a data perspective. However, the correlation investigated in our work is more about the translation of RM errors into policy regret.
>
>
>     | Experiment | Kendall corr. | Pearson corr. | Spearman corr. | MRR |
>     | --- | --- | --- | --- | --- |
>     | BoN / Test Acc. | 0.6561 | 0.7520 | 0.7533 | 0.6333 |
>     | PPO / Test Acc. | 0.4654 | 0.6395 | 0.6102 | 0.5167 |
>     | Train Acc. / Test Acc. | 0.6393 | 0.8952 | 0.7207 | 0.5301 |

---

> ### Author Response · Authors · 2024-11-23
>
> *Responses to Ablations:*
>
> 1. **Regarding the KL penalty**, we set it to 0, following the experimental setup from [1]. However, we acknowledge that this is a point worth investigating. To address this, we included additional experiments in Appendix 8.5 to observe the impact of the KL penalty on correlation. These experiments reveal that while appropriately increasing the KL penalty can enhance the Spearman correlation, it may reduce the MRR. Conversely, when the KL penalty becomes too large, it negatively affects the Spearman correlation. Our findings suggest that a smaller KL penalty allows for a more stable and localized PPO optimization process, thus improving the predictive correlation of RM error. However, increased KL penalties can limit the expected reward's growth; when too high, they may even cause training instability. In our experiments, we found that a KL penalty of 0.5 can lead to a training collapse.
>     - [1] OpenAI. Scaling Laws for Reward Model Overoptimization.
> 2. **Regarding the points with low accuracy but high NDR**, we found that these occur when two specific RMs are used as the Golden RM. These RMs were trained on datasets with 10% and 45% noise, respectively. Upon examining their RL training processes, we discovered that, compared to other RMs, the optimization for these two involved the least KL divergence (resulting in a token KL of about 0.8, whereas other RMs were around 0.15). For the RM trained with 45% data noise, we believe this phenomenon is understandable. Given the higher noise in the training data, the RM signals are likely noisier, making them more challenging to optimize. However, overall, it shares similar preferences with other RMs (since all were trained on the same dataset with varying noise levels). Thus, optimizing toward RMs with less noise could enhance the expected rewards of this noisier RM, while also achieving a greater KL divergence. As for the RM trained with 10% data noise, we think the characteristics of the RM itself or the PPO optimization process might be contributing factors. We observed that while this RM's KL increased rapidly during the initial phase of training, it slowed down later, potentially indicating that it became trapped in a local optimum. We believe that analyzing the RL process and gaining a deeper understanding of the inherent properties of RMs can help us better predict these outliers in future work.
> 3. **Regarding whether adding more samples in Figure 6** would further improve correlation or reach saturation, we addressed this concern by examining how correlation changes with increasing sample sizes, particularly beyond 5000, as discussed in Appendix 8.6. As depicted in Figure 12, we found that further expanding the sample size does not reduce correlation. Instead, the correlation gradually approaches an upper limit, indicating saturation.
> 4. **Regarding the question of whether the downstream policy affects the benefits of adding more responses per prompt**, we conducted further analysis, detailed in Appendix 8.7. In this experiment, we used Qwen-2.5-7B-Instruct as the downstream policy instead of LLaMA-3-8b-Instruct to determine whether this strategy remains beneficial. As shown in Figure 15, the advantage persists. We hope this addresses your concern.
> 5. **Regarding the differences between the prompts of RM and RL test datasets**, we acknowledge the importance of your concern. In some real-world scenarios, the prompts in the RM test set and the RL test set may differ. For example, RM performance might be tested on RewardBench while RL-trained policy performance is assessed on other downstream tasks. Therefore, we explored how this discrepancy affects the correlation between RM test metrics and downstream performance in Finding 3. Our findings indicate that differing prompts potentially weaken the correlation, particularly with PPO. We hope this addresses your question.

---

> > ### Author Response · Authors · 2024-11-23
> >
> > *Response to Notations and Clarity:*
> >
> > 1. **Notations:** We have corrected the notation errors and added supplementary descriptions in lines L144 to L154. In response to your questions, $\pi_0$ indeed represents the initial policy. Additionally, we have added a citation for the regularization term.
> > 2. **Clarity:** We greatly thank you for your suggestion. We have made corresponding improvements in the lines that you pointed out and other places that we can find.
> >     1. In response to the question regarding "the difficulty of constantly controlling optimization pressure," we have revised the description of optimization pressure and added a formal definition in Equation (2). We used the term "optimization pressure" to denote the degree of optimization towards a particular RM, now defined directly using KL divergence. The challenge in controlling it arises from the difficulty in ensuring that all trained policies maintain a similar range of KL divergence. For instance, in PPO, the KL divergence between the optimized policy and the initial policy may vary due to differences in learning rates, KL penalties, and other factors.
> >     2. In response to the confusing sentence, “Can we achieve a higher correlation by sampling responses specifically from the model used for downstream optimization? To examine this question, we construct test datasets with responses solely from different models,” we intended to question whether it is necessary to sample responses from the model used for downstream optimization to construct the test dataset. To investigate this question, we constructed multiple test datasets. Each test dataset contains responses sampled exclusively from a single model. We have revised the content to clarify this point.
> > 3. We greatly appreciate your feedback on the writing issues. We have revised the relevant sections and made every effort to address other potential problems.
> >
> > *Response to the questions:*
> >
> > 1. **Regarding the high correlation between accuracy and BoN perf at high accuracy**, we observed that these generally occur when RMs trained on datasets with less noise are paired as the golden-proxy. We believe this indicates that when training data is less noisy, the RM signals are clearer. Consequently, the accuracy obtained on the test set can better reflect the preference consistency between RMs, leading to better predictions of downstream performance.
> > 2. **Regarding the question about the sampling model**, we believe that as the RL optimization process progresses, the distribution of responses increasingly deviates from the initial model distribution. Therefore, sampling responses solely from the initial distribution may not adequately predict the performance of the optimized policy distribution. Conversely, as discussed in Finding 2, the rank distribution of responses in the test set may have a more significant impact on correlation. The consistency between the Golden and Proxy RMs on higher-ranked responses may better reflect their influence on the optimized policy, which tends to generate responses that are ranked higher.

---

> ### Author Response · Authors · 2024-11-24
>
> Dear Reviewer,
>
> We sincerely appreciate your detailed and insightful comments, as well as the time and effort you've dedicated to reviewing our work. We have made every effort to supplement the experiments and address your concerns. With the rebuttal deadline approaching, we look forward to any further feedback you might have. Please feel free to reach out with any additional questions or concerns.

---

> > ### Comment · Reviewer_22rE · 2024-11-25
> > **Follow-up [22rE]**
> >
> > I thank authors for the effort they put in answering my main concerns. With the revisions in, I think the paper will be in a much better state. I would advise authors to integrate citations to concurrent work (RMB, RewardMath) in the revised version. I raised my score accordingly.

---

### Official Review · Reviewer_Rn3n · 2024-11-04

**Soundness:** 4
**Presentation:** 3
**Contribution:** 3
**Rating:** 8
**Confidence:** 3

**Summary:**

Research question of the paper is on how we should evaluate the quality of the reward models for RLHF. The paper conducts experiments to evaluate the evaluation metrics for RMs and show interesting findings that some confirm the intuition and the other are somewhat counterintuitive. The paper concludes that there should be more care on evaluating RMs instead of relying on a single benchmark.

**Strengths:**

Overall, the paper presents empirical results that evaluate the metrics of RMs, which the community has previously intuited but not with rigorous scientific evaluation. The experiments are designed to test hypotheses incrementally, helping the RLHF community build a comprehensive body of knowledge on RM evaluation.

- The paper tackles the question that the alignment research community needs to know the answer to.
- Table 2 is interesting as it shows counterintuitive results. One would guess that the RM should be evaluated for the samples generated from the policy to be trained. The paper also shows that instead, we should sample multiple responses and choose a pair of responses to evaluate the accuracy with care.

**Weaknesses:**

I don't see any critical weaknesses for the paper. If I were to come up with the weaknesses:

- Although policy regret is often referred to in the paper, its formal definition is not clearly stated. It would be better to have an equation defining the regret. Even if we do not have a way to compute it, the goal of the research is to estimate it so I would say that it is worth clarifying its definition formally.
- The scope of the paper is to show that the current evaluation scheme is not enough (which is a good enough contribution). The paper does not provide a solution to the problem of how we should evaluate the RMs (which I think is asking too much).
- The texts in Figures are a bit too small to read. It would be nice if it is a bit larger.

**Questions:**

- Randomly flipping some percent of the trains in the training dataset is a trick to make a pseudo proxy reward model used in several papers (e.g., AlpacaFarm; Dubois+ 2024). In reality, it is more natural to think that some kinds of instructions have more flips and others have less. I'm a bit concerned that the findings of the paper might only hold true due to the synthetic error model. It would be helpful to see whether this possibility is addressed or refuted (or let me know if I missed it).

---

> ### Author Response · Authors · 2024-11-23
>
> Thank you for the positive feedback and the constructive comments. We address your concerns below.
>
> *Response for Weakness*
>
> **W1: Formal definition of policy regret**
>
> We revised lines L144-L154 to clarify the definition of regret in the context of RLHF in the latest version of our paper. Specifically, given that the KL divergence between a policy $\pi$ and the initial policy $\pi_0$ is $KL(\pi||\pi_0)=\lambda$, the regret with respect to a golden reward function  $r^*$ is defined as follows:
>
> $$
> Reg_{r^\ast} = \frac{\max\limits_{KL(\pi' \parallel \pi_0)<\lambda} J_{r^\ast}(\pi') - J_{r^\ast}(\pi)}
> {\max\limits_{KL(\pi' \parallel \pi_0)<\lambda} J_{r^\ast}(\pi') - \min\limits_{KL(\pi' \parallel \pi_0)<\lambda} J_{r^\ast}(\pi')}
> $$
>
>
> This reflects the ratio of the maximum possible reward gain to the actual reward gain obtained, given the KL divergence constraint. We added detailed descriptions for relevant concepts that might potentially cause confusion.
>
> **W2: Solution of how we should evaluate the RMs**
>
> Our primary focus is on highlighting the potential limitations of current RM evaluation methods. Regarding better RM evaluation, we explored factors that could influence the correlation between these metrics and downstream performance in Section 4. These findings can aid in the development of RM benchmarks that are more closely aligned with downstream tasks.  For future research, we consider RM interpretability a promising direction for enhancing our understanding of RM evaluation, which could help better predict various Goodhart’s effects beyond the Regressional Goodhart's effects discussed in this paper.
>
> **W3: Texts size in Figures**
>
> Sorry for the inconvenience. We have modified Figure 3 to improve its legibility.
>
> *Response the question:*
>
> Thank you for raising this concern. We adopt this setting in line with previous works [1,2,3]. We believe this approach reflects the data noise present in training datasets, which is also a common issue in real-world scenarios.
>
> - [1] Impact of Preference Noise on the Alignment Performance of Generative Language Models
> - [2] AlpacaFarm: A Simulation Framework for Methods that Learn from Human Feedback
> - [3] B-Pref: Benchmarking Preference-Based Reinforcement Learning

---

> > ### Comment · Reviewer_Rn3n · 2024-11-25
> >
> > Thank you very much for the response.
> >
> > I think with the clarification of the policy regret the paper is in really good shape. I don't see a reason to reject it.

---

### Author Response · Authors · 2024-11-23

We sincerely thank all the reviewers for their valuable and thorough feedback on our work. We individually respond to each reviewer's comments and have incorporated common suggestions, enriching our paper with additional results. Below, we outline the major changes:

- In Section 1, we revised some overly absolute statements and amended expressions that could be confusing without the full context of the paper.
- In Section 2, we added the definition of policy regret within the context of RLHF and clarified related concepts. Additionally, we included information on label consistency between datasets and cited the term for RM loss regularization.
- In Section 3, we included an explanation at the beginning regarding the use of RewardBench data and re-drew Figure 3 for improved readability.
- In Section 4, we revised content and presentation details to prevent comprehension difficulties.
    - For Finding 2, we revised explanations regarding the necessity of sampling from the downstream model.
    - For the rank experiment shown in Figure 4, we added axis labels, included detailed explanations in the caption, and provided a more thorough analysis.
    - Regarding the annotation budget in Finding 4, we enhanced the descriptions of the results presented in Figure 7.
- In Section 5, we refined the insights to enhance clarity and added accuracy information to Figure 9 for reference.
- In Appendix 8.3, we included accuracy details for Golden-Proxy pairs and information on policy regret under BoN and PPO.
- In Appendix 8.4, we added correlation results under various synthetic RM settings (different model scales and training sets) and examined the correlation between diverse RM performances (RewardBench) and downstream tasks (MT-bench) in real-world scenarios.
- In Appendix 8.5, we supplemented our study with ablation experiments on the KL Penalty in PPO training to explore its impact.
- In Appendix 8.6, we discussed changes in correlation with further expansion of sample size.
- In Appendix 8.7, we validated the effectiveness of increasing the number of responses per prompt when using different downstream models.
- In Appendix 8.9, we further introduced the definition and derivation of the degree of overoptimization.
- In Appendix 8.10, we detailed the experiments, explaining how multiple samplings were conducted for more reliable results, and provided specific parameters for BoN and PPO training.

---

### Meta-Review · Area_Chair_SU5h · 2024-12-21

**Metareview:**

This paper questions the current evaluation procedure of reward models in LLM post-training. It shows that the accuracy of reward models does not necessarily translate to the improvement in downstream RLHF tasks.

Strengths:
This paper studies an important and well-motivated problem — reward modeling is important in LLM post-training like RLHF, but hasn't been deeply investigated. This paper presents several interesting observations through comprehensive experiments.

Weaknesses:
The reviewers have concerns about weaknesses around paper presentation and ablation studies. Most of the concerns were addressed during the rebuttal period.

All reviewers agree that this paper studies an important problem and obtains interesting observations. I agree with this assessment and believe this paper would be a good addition to the community.

**Additional Comments On Reviewer Discussion:**

The concerns from the reviewers are mostly around writing and ablation studies. The authors provided additional experiments and addressed the issues on ablation studies, as well as explanation to address most of the concerns on writing.

---

### Decision · Program_Chairs · 2025-01-22

Accept (Spotlight)